# Development of a Laser-Photofragmentation Laser-Induced Fluorescence instrument for the detection of nitrous acid and hydroxyl radicals in the atmosphere

Brandon Bottorff[1], Emily Reidy[1], Levi Mielke[2,a], Sebastien Dusanter[2,b], and Philip S. Stevens[1,2]

[1]Department of Chemistry, Indiana University, Bloomington, IN, 47405, USA
[2]O'Neill School of Public and Environmental Affairs, Indiana University, Bloomington, IN, 47405, USA
[a]now at Department of Chemistry, University of Indianapolis, Indianapolis, IN, 46227, USA
[b] now at IMT Lille Douai, Institut Mines -Télécom, Univ. Lille, Centre for Energy and Environment, F-59000 Lille, France

*Correspondence to*: Philip S. Stevens (pstevens@indiana.edu)

**Abstract.** A new instrument for the measurement of atmospheric nitrous acid (HONO) and hydroxyl radicals (OH) has been developed using laser photofragmentation (LP) of HONO at 355 nm after expansion into a low-pressure cell, followed by resonant laser-induced fluorescence (LIF) of the resulting OH radical fragment at 308 nm similar to the fluorescence assay by gas expansion technique (FAGE). The LP/LIF instrument is calibrated by determining the photofragmentation efficiency of HONO and calibrating the instrument sensitivity for detection of the OH fragment. In this method, a known concentration of OH from the photo-dissociation of water vapor is titrated with nitric oxide to produce a known concentration of HONO. Measurement of the concentration of the OH radical fragment relative to the concentration of HONO provides a measurement of the photofragmentation efficiency. The LP/LIF instrument has demonstrated a 1σ detection limit for HONO of 9 ppt for a 10-min integration time. Ambient measurements of HONO and OH from a forested environment and an urban setting are presented along with indoor measurements to demonstrate the performance of the instrument.

## 1 Introduction

Although the photolysis of ozone followed by the reaction of excited oxygen atoms with water vapor has been recognized as an important source of hydroxyl radicals (OH) in the troposphere (Rohrer and Berresheim, 2006), several studies have indicated that the photolysis of nitrous acid (HONO) (R1) is a significant, if not dominant source of OH in several environments (Kleffmann et al., 2005; Acker et al., 2006; Dusanter et al., 2009b; Volkamer et al., 2010; Ren et al., 2013; Griffith et al., 2016).

$$\text{HONO} + h\nu \ (300 \text{ nm} < \lambda < 400 \text{ nm}) \ \rightarrow \text{OH} + \text{NO} \quad\quad\quad\quad\quad\quad \text{(R1)}$$

As the dominant oxidant in the lower troposphere, OH initiates reactions with carbon monoxide and a wide variety of volatile organic compounds (VOCs) leading to the formation of the hydroperoxy radical ($HO_2$) and organic peroxy radicals ($RO_2$). In the presence of nitrogen oxides ($NO_x = NO + NO_2$) reactions of these peroxy radicals regenerate OH radicals, establishing a fast radical-propagation chain that can produce harmful pollutants including ozone and secondary organic aerosols. Attempts to develop effective control strategies for these secondary pollutants necessitate a thorough understanding of OH radical chemistry. Due to its importance as a radical precursor, a more complete understanding of HONO sources and sinks is critical to understanding the oxidative capacity of the atmosphere.

HONO is produced in the gas phase from the reaction of OH radicals with NO (R2). In addition to photolysis (R1), reaction of HONO with OH radicals (R3) is another important loss mechanism in the gas phase. Considering R2 as the only source of HONO, its gas-phase concentration can be calculated from a steady-state expression (Eq. 1).

$$OH + NO + M \rightarrow HONO + M \qquad\qquad (R2)$$

$$HONO + OH \rightarrow H_2O + NO_2 \qquad\qquad (R3)$$

$$[HONO]_{PSS} = \frac{k_{OH+NO}[OH][NO]}{J_{HONO}+k_{OH+HONO}[OH]} \qquad\qquad (1)$$

In this equation, $k_{OH+NO}$ is the rate constant for reaction R2, $J_{HONO}$ is the photolysis rate constant for reaction R1, and $k_{OH+HONO}$ is the rate constant for reaction R3. Compared to other photolytic sources of OH, the longer wavelengths at which HONO photolyzes to produce OH can result in HONO photolysis dominating OH production during the morning hours in some environments (Volkamer et al., 2010), but decrease in importance during the day as the concentration of HONO decreases. However, in several instances HONO photolysis has been shown to be a significant OH source through the day (Kleffmann et al., 2005; Acker et al., 2006; Ren et al., 2013; Griffith et al., 2016; Xue et al., 2020). This is mainly due to higher-than-expected daytime HONO mixing ratios that cannot be attributed to gas-phase reactions (R1-R3) (Tang et al., 2015; Lee et al., 2016; Meusel et al., 2016; Xue et al., 2020).

Other HONO sources include direct emission from vehicles or other combustion sources (Kirchstetter et al., 1996; Kurtenbach et al., 2001; Li et al., 2008; Xu et al., 2015), direct photolysis of some species (Bejan et al., 2006; Zhou et al., 2011), photo-enhanced surface reactions (George et al., 2005; Stemmler et al., 2006), and release from soil due to biological processes (Su et al., 2011; Oswald et al., 2013; Weber et al., 2015; Meusel et al., 2018). Lastly, several production pathways involving the heterogeneous conversion of $NO_2$ to HONO on soil, leaf canopies, aerosols, and other surfaces have been suggested to explain higher-than-expected HONO mixing ratios observed during some field campaigns (Kleffmann et al., 1998; Ramazan et al., 2004; Stutz et al., 2004; Xue et al., 2020).

HONO is also an important pollutant within the indoor environment. While outdoor mixing ratios during the daytime are typically within the range of tens to hundreds of ppt (Huang et al., 2002; Oswald et al., 2015), and can range from hundreds of ppt up to several ppb at night or during morning rush hour in urban environments (Stutz et al., 2010; Young et al., 2012; Xu et al., 2015; Lee et al., 2016), indoor HONO measurements have shown background levels of several ppb, and elevated mixing ratios as high as 20-90 ppb during cooking or other combustion events (Brauer et al., 1990; Vecera and Dasgupta, 1994; Zhou et al., 2018; Liu et al., 2019; Wang et al., 2020a). In two studies, average outdoor mixing ratios were 0.9 and 0.3 ppb compared to 4.6 and 4.0 ppb in nearby suburban homes (Leaderer et al., 1999; Lee et al., 2002). Elevated concentrations of HONO indoors are relevant not only due to the adverse health effects caused by inhalation (Beckett et al., 1995; van Strien et al., 2004; Jarvis et al., 2005), but also due to the potential for OH production indoors from HONO photolysis. OH concentrations were thought to be negligible indoors due to reduced light intensity, especially at short wavelengths and lower ozone mixing ratios, but several studies have indicated that photolysis of elevated indoor HONO can produce OH concentrations similar to those found outdoors, even at reduced photolysis frequencies (Gómez Alvarez et al., 2013; Bartolomei et al., 2015; Kowal et al., 2017).

As a result of these observations, a clear understanding of HONO sources is an important step in understanding the overall oxidation capacity of both the outdoor and the indoor environments. However, detailed mechanisms and dependence on variables such as surface type and chemical composition are still lacking for both heterogeneous HONO sources and photo-enhanced surface reactions. Thus, additional measurements of HONO in various environments and from laboratory experiments are still needed for the development of a more complete understanding of both HONO formation mechanisms and its potential to initiate the radical chain that leads to secondary pollutant formation.

Several different measurement techniques have been employed to measure HONO, beginning with Differential Optical Absorption Spectroscopy (DOAS) (Perner and Platt, 1979). DOAS is based on the UV-visible absorption of HONO in the atmosphere across path-lengths of several kilometers and provides a direct measurement that does not require external calibration. The open-path

nature of DOAS also eliminates potential impacts from inlet or surface chemistry that could result in interferences or loss of HONO, but the long path length required also limits its spatial resolution (Tsai et al., 2018). Incoherent broadband cavity-enhanced absorption

spectroscopy (IBBCEAS) is another optical technique that is capable of measuring HONO and several other trace gases. A long path length, similar to that used in DOAS instruments, is maintained within a short cavity of 0.5-2.0 m using two highly reflective mirrors (Nakashima and Sadanaga, 2017; Jordan and Osthoff, 2020; Tang et al., 2020). Detection limits from IBBCEAS have improved in recent years to as low as 118 ppt for a 60s integration time, but still may not be sufficient for ambient measurements in less polluted environments (Duan et al., 2018).

Several wet chemical techniques are also capable of detecting HONO, including but not limited to, wet denuder-ion chromatography (IC) (Neftel et al., 1996), mist chamber-IC (Dibb et al., 2004), 2,4-dinitrophenylhydrazone derivatization/high-performance liquid chromatography (DNPH derivatization/HPLC) (Zhou et al., 1999), derivatization with sulfanilamide/N-(1-naphthyl)-ethylenediamine/high-performance liquid chromatography (Afif et al., 2016), and long optical path absorption photometry (LOPAP) (Heland et al., 2001). These techniques offer low detection limits and integration times, often below 5 ppt and a few minutes

respectively, but indirectly measure gaseous HONO by conversion to nitrite anion or a dye within a liquid solution. This conversion introduces the potential for sampling artifacts or chemical interferences where other species may also be converted and interpreted as HONO. For example, measurements of HONO using LOPAP have been shown to have high sensitivity and limits of detection less than 1 ppt but suffer from interferences from atmospheric concentrations of $NO_2$ and $O_3$ (Heland et al., 2001). In addition, peroxyacetyl nitrate (PAN) and peroxynitric acid ($HO_2NO_2$) can be partially observed as HONO in these instruments (Villena et al., 2011; Legrand

et al., 2014). LOPAP instruments typically utilize two stripping coils connected in series to correct for these and other unknown interferences. In the first coil, HONO is trapped efficiently along with some interfering species. These interferences are similarly trapped in the second coil, which allows a true HONO signal to be determined by subtraction (Heland et al., 2001; Legrand et al., 2014). More recently, Chemical ionization Mass spectrometry (CIMS) has been used to measure HONO along with other inorganic acids. Iodide ion ($I^-$) and acetate ion ($CH_3COO^-$) CIMS have both been used with reported detection limits of 30 ppt (Roberts et al., 2010; Veres et al.,

2015; Collins et al., 2018).

Other methods to measure ambient HONO include laser-photolysis into OH and NO fragments and subsequent detection of OH by laser-induced fluorescence at atmospheric pressure (Liao et al., 2006a). This method was used successfully to detect ambient HONO at the South Pole in 2003 (Liao et al., 2006b). Although this instrument exhibited a low detection limit of 15 ppt for a 1-min integration time, the wavelength of 282 nm used for excitation of OH made it less suitable for environments with higher ozone and water

vapor mixing ratios due to the potential for laser-generated OH inside the detection cell from the photolysis of ozone followed by reaction of $O(^1D)$ with water vapor (R4 and R5) (Wennberg et al., 1994). This interference can impact the detection limit of HONO by increasing the measured OH background signal.

$$O_3 + h\nu \ (\lambda < 340 \ nm) \ \rightarrow O(^1D) + \ O_2 \hspace{5cm} (R4)$$

$$O(^1D) + H_2O \ \rightarrow 2OH \hspace{7cm} (R5)$$

More recently, Dyson et al. (2021) report the detection of HONO in a laboratory setting using laser-photolysis of HONO at 355 nm and subsequent detection of OH at 308 nm in a low-pressure detection cell, reporting a detection limit of 12 ppt for a 50-s average.

Despite the importance of measuring HONO in the atmosphere, recent instrument intercomparisons have revealed significant discrepancies in measurements of HONO between various instrumental techniques (Pinto et al., 2014; Ródenas et al., 2013; Crilley et al., 2019). In this paper we describe a new laser-photofragmentation/laser-induced fluorescence instrument capable of near simultaneous

measurement of both HONO and OH. In this approach, photofragmentation of HONO and detection of the OH fragment occur after

sampling ambient air at low pressure, similar to the fluorescence assay by gas expansion (FAGE) technique currently used to measure ambient concentrations of the OH radical in the atmosphere (Heard and Pilling, 2003). Excitation and detection of OH occurs at 308 nm as this wavelength is much less susceptible to interference from laser-generated OH from reactions R4 and R5 because the ozone absorption cross section is only 4% of that at 282 nm (Heard and Pilling, 2003; Burkholder et al., 2019) and sampling at low pressure reduces the concentration of ozone and water vapor in the detection cell. In addition to a description of the instrument, a calibration method for HONO based on a measurement of the photofragmentation efficiency is described, and examples of measurements of HONO concentrations by this instrument in both outdoor and indoor environments are presented.

## 2 Experimental Section

### 2.1 Instrument description

The Indiana University laser-photofragmentation/laser-induced fluorescence (LP/LIF) instrument consists of four primary components: (1) a photolysis laser that fragments HONO into OH and NO, (2) a 308-nm laser for the excitation of the OH radicals, (3) a low-pressure sampling cell, and (4) a sensitive gated detection system that synchronizes photofragmentation of HONO, excitation of OH, and detection of the resulting OH fluorescence. A schematic of the instrumental configuration is shown in Fig. 1.

The absorption spectrum of HONO is shown in Fig. S1 (Burkholder et al., 2019). The strong peak near 355 nm was chosen for photofragmentation because it coincides with the third harmonic of a Nd:YAG laser. A Spectra Physics Navigator II YHP40-355HM laser is used for photofragmentation of HONO, producing approximately 3-4 W of radiation at 355-nm and at a repetition rate of 10 kHz with a pulse width of approximately 20 ns. The OH excitation laser system consists of a Spectra Physics Navigator II YHP40-532 Nd:YAG laser that produces approximately 7-8 W of radiation at 532-nm at a repetition rate of 10 kHz and a pulse width of approximately 20 ns. This laser pumps a dye laser (Sirah Credo, 255 mg $L^{-1}$ of Rhodamine 610 and 80 mg $L^{-1}$ of Rhodamine 101 in ethanol) to produce approximately 40-100 mW of radiation at 308 nm. A small portion of the 308-nm emission is diverted to a low-pressure reference cell for wavelength calibration (Dusanter et al., 2009a). In this cell, a high concentration of OH radicals is produced by the thermal dissociation of water vapor using a hot alumel filament. The resulting OH fluorescence is collected by a Hamamatsu photomultiplier tube (H6180-01) equipped with a bandpass filter centered at 308 nm (ESCO products). Using the OH fluorescence signal from this cell, the excitation laser is tuned to the $Q_1(3)$ transition of OH at 308.1541 nm, a transition that exhibits one of the strongest absorption cross sections around 308 nm (Dusanter et al., 2009a).

The sampling cell is shown in Fig. 2. Ambient air is drawn into the sampling cell through a flat 1-mm diameter pinhole inlet by means of two scroll pumps (Edwards XDS35i) connected in parallel. The cell is maintained at a pressure of 0.25 kPa to reduce quenching of the OH fluorescence by ambient air, and thus increase the OH radical fluorescence lifetime. As the sampled air passes through the inlet, it expands into a central aluminum cube and is intersected by the fragmentation and excitation laser emissions. The 355-nm laser emission is propagated to the sampling cell through a 12-m long, 1000-micron fiber-optic patch cord (Oz Optics) which results in approximately 1.5 W of laser power at the entrance of the sampling cell. The excitation laser emission is propagated to the cell through a 12-m long, 200-micron fiber-optic cable (Thorlabs, FG200AEA) which results in approximately 1-4 mW of 308-nm radiation into the sampling cell. After exiting their respective fiber-optic cables, both the 355-nm and 308-nm laser emissions are spatially combined by a dichroic mirror (Rocky Mountain Instrument Co.) that reflects greater that 90% of the 308-nm laser beam and transmits greater than 90% of the 355-nm laser beam. The beams are temporally separated, with the 308-nm pulse entering the detection cell 100 ns after the 355-nm pulse. Upon exiting the detection cell, the beams are spatially separated by a second dichroic mirror, and the power of each beam is monitored using a photodiode (UDT-555UV, OSI Optoelectronics) equipped with interference filters (Esco Optics, Thorlabs).

The fluorescence from the OH radical fragment is collected by an optical train orthogonal to the excitation beam. Two lenses (f=75 mm, CVI Laser) focus the fluorescence onto the detector, and a band-pass filter centered at 308 nm (Barr Associates, transmission 65%, bandwidth 5 nm, OD>5 at other wavelengths) selectively passes OH fluorescence to the detector and reduces the detection of solar scatter, potential broadband fluorescence of other species, and scatter from the 355-nm laser. A concave mirror (100-mm diameter, 40-cm focal length, Melles Griot) opposite the optical train effectively doubles the solid angle of collection.

The detection system consists of a time-gated micro-channel plate photomultiplier tube (MCP-PMT) (Photek PMT325), a preamplifier/discriminator (F-100T, Advanced Research Instruments) and a high-speed photon counter (National Instruments, 6024E). A delay generator (Berkley Nucleonics Model 575) triggers both laser emissions, separated by 100 ns, and also increases the MCP-PMT gain after the 308-nm laser pulse (Fig. 3). Turning on the detector after the 308-nm laser pulse reduces the detection of the intense scattered radiation from the laser pulse while allowing the detection of the OH fluorescence. The gain of the detector is reduced during the laser pulse and switched to the highest gain approximately 70-ns after the laser pulse. The gain is kept high for approximately 550 ns in order to collect most of the OH fluorescence, and then is reduced until the next laser pulse. The signal from the MCP is amplified and filtered by a pulse-height discriminator (F100T) that delivers TTL pulses for each detected photon. The photon counter is set with a timing gate to count the fluorescence photons during a 400-ns window while the gain of the detector is high, avoiding potential noise associated with the increase and decrease in detector gain (Fig. 3).

Wavelength modulation is used to tune the 308-nm dye laser excitation emission on- and off-resonance with the $Q_1(3)$ transition of OH at 308.1451 nm. The net signal from the OH fluorescence is derived by subtracting the off-resonance signal, consisting primarily of solar scatter and some scattered laser radiation that extends into the detection window, from the on-resonance signal. To differentiate OH fluorescence signals due to HONO photofragmentation from fluorescence due to ambient OH radicals, the 355-nm fragmentation laser is cycled on and off with the use of a diaphragm shutter (Thorlabs). When combined with dye-laser wavelength modulation, this creates a complete measurement sequence that allows near simultaneous measurement of both ambient OH and HONO. Figure 4 shows an example of a typical measurement sequence. Each measurement cycle consists of four 15-s steps – (1) a background signal is established where HONO is photolyzed but the 308-nm laser is tuned off-resonance ($S_1$), (2) both ambient OH and the OH fragment from HONO are excited by tuning the 308 nm laser to on resonance ($S_2$), (3) the 355-nm photolysis laser is blocked by a shutter and background signal is re-established by tuning the 308-nm laser off-resonance ($S_3$), and (4) the 355-nm laser is still blocked but ambient OH is excited by tuning the 308-nm laser on-resonance ($S_4$). The net HONO signal is obtained from the difference between the signals from cycles 2 and 4 ($Net_{HONO} = S_2 - S_4$), while the net ambient OH signal ($Net_{OH}$) is obtained from the difference between the signals from cycles 4 and the background signal ($S_{bkg}$) which is the average of cycles 3 and 1 ($Net_{OH} = S_4 - S_{bkg}$). Because the band-pass filter rejects scatter from the 355-nm laser, the background signal with and without the fragmentation laser are not significantly different and only varies with fluctuations in the 308-nm laser power.

In addition to the sequence described above, chemical modulation cycles can be used to test for potential interferences in the measurement of ambient OH as performed on FAGE instruments (Mao et al., 2012; Novelli et al., 2014; Rickly and Stevens, 2018). Due to the single-pass laser design of the LP/LIF sampling cell, signals due to laser generated OH from reactions R4 and R5 are small and are calibrated as a function of laser power, ozone and water concentrations (Griffith et al., 2016). However, Criegee intermediates formed from the ozonolysis of alkenes (Rickly and Stevens, 2018), the decomposition of ROOOH species (Fittschen et al., 2019), or other unknown interferences, could lead to the formation of OH radicals inside the detection cell. Removal of ambient OH through external reaction with a scrubbing agent, such as perfluorpropylene ($C_3F_6$) (Griffith et al., 2016; Rickly and Stevens, 2018), allows quantification of all OH formed within the detection cell. The remaining OH signal would be a measurement of the interference that can be subtracted from the total OH signal when the ambient OH is not removed. While the LP/LIF instrument can incorporate chemical

modulation cycles to measure potential interferences, the technique was not used in the HONO and OH measurements reported below in order to increase the HONO measurement frequency.

The LP/LIF instrument is automated using a National Instruments multifunction DAQ board (NI USB 6024) and a customized LabView interface program that controls the 355-nm laser shutter and monitors the power of both the 355-nm fragmentation laser and the 308-nm excitation laser. The 308-nm laser is controlled by the Indiana University Laser-Induced Fluorescence – Fluorescence Assay by Gas Expansion (IU-FAGE) instrument as described previously (Dusanter et al., 2009a). The output of the 308-nm laser system is split between the LP/LIF instrument and the IU-FAGE instrument, allowing simultaneous measurements of ambient HONO and OH by

the LP/LIF instrument with measurements of ambient concentrations of OH and $HO_2$ by the IU-FAGE instrument (Dusanter et al., 2009a).

## 2.2 Instrument calibration

Several of the previously mentioned measurement techniques utilize a HONO generation source to characterize instrumental response to a known concentration of HONO. Many sources are based on the design of Febo et al. (1995) in which gaseous hydrochloric acid

reacts with sodium nitrite to form HONO (R6):

$$HCl(g) + NaNO_2(s) \longrightarrow HONO(g) + NaCl(s) \tag{R6}$$

This method requires the reaction chamber to be heated to 50°C and can produce mixing ratios of HONO over a wide range (5-20000 ppb) (Febo et al., 1995), that often require large dilution flows to reach typical outdoor atmospheric concentrations (Lao et al., 2020). Furthermore, the high HONO mixing ratios produced by this approach can disproportionate to form NO and $NO_2$ (Febo et al., 1995;

Stutz et al., 2000; Gingerysty and Osthoff, 2020; Lao et al., 2020). As a result, this method typically requires an additional technique to verify both the purity and output concentration of HONO (Pérez et al., 2007; Gingerysty and Osthoff, 2020). While appropriate for a laboratory setting, these limitations, along with the long warmup times needed to ensure stability, can make this method less suitable for calibration in a field setting.

        Instead, the sensitivity of the LP/LIF instrument to HONO ($R_{HONO}$) is determined from (1) the photofragmentation efficiency

of HONO by the 355-nm laser (PE), and (2) the sensitivity of the instrument to the measurement of the OH fragment ($R_{OH}$):

$$R_{HONO} = R_{OH} \times PE \tag{2}$$

The instrumental sensitivity towards OH ($R_{OH}$) is determined using the water-vapor photolysis technique which has previously been described in detail (Dusanter et al., 2008). Briefly, this method relies on the photolysis of water vapor at 184.9 nm to produce a known amount of OH (and $HO_2$):

$$H_2O + h\nu \text{ (184.9 nm)} \longrightarrow OH + H \tag{R7}$$

$$H + O_2 + M \longrightarrow HO_2 + M \tag{R8}$$

$$[OH] = [HO_2] = [H_2O] \times \sigma_{H_2O} \times \varphi_{OH+H} \times (F \times t) \tag{3}$$

As shown in Eq. (3), the concentration of OH and $HO_2$ produced by the calibrator can be calculated from the time-integrated photolysis of water vapor. In this equation, $\sigma_{H2O}$ is the absorption cross-section of water at 184.9 nm ($6.78\times10^{-20}$ $cm^2$ $molecule^{-1}$) and $\varphi_{OH+H}$ is the

unity photo-dissociation quantum yield (Burkholder et al., 2019). $F$ is the photon flux and $t$ is the photolysis exposure time. The quantity $(F \times t)$ can be determined via $O_2$ actinometry experiments, as molecular oxygen is also photolyzed at 184.9 nm to form $O^3(P)$ and then $O_3$ after reaction with $O_2$. The concentration of ozone produced is also dependent on the product of $(F \times t)$:

$$[O_3] = [O_2] \times \sigma_{O_2} \times \varphi_{O_3} \times (F \times t) \tag{4}$$

$$(F \times t) = \frac{[O_3]}{2 \times [O_2] \times \sigma_{O_2}} \qquad (5)$$

In these equations, $\varphi_{O3}$ is the quantum yield of ozone from oxygen photolysis ($\varphi_{O3} = 2$) and $\sigma_{O2}$ is the effective absorption cross-section of oxygen at 184.9 nm. Thus, measurements of ozone concentrations can be used to determine the quantity $(F \times t)$ if the effective oxygen absorption cross section is known. Previous studies have shown that the effective oxygen absorption cross section at 184.9 nm is dependent on operating conditions ($O_2$ column density, lamp current, and lamp temperature), making it necessary to measure $\sigma_{O2}$ for

each calibration system (Hofzumahaus et al., 1997; Lanzendorf et al., 1997). The dependence of the effective oxygen absorption cross section on the mercury lamp is the result of the overlap between several features of the highly structured Schumann-Runge band and the lamp emission at 184.9 nm that depends on the operating conditions due to line reversal (Lanzendorf et al., 1997) and potential fluorescence of the fused silica envelope (Cantrell et al., 1997).

     Using Eq. (3-5), a known concentration of OH can be produced from known concentrations of water vapor and ozone, and the instrumental sensitivity towards OH (Eq. 6) can be derived from the measured fluorescence signal ($S_{OH}$) and normalized to the power

of the 308-nm laser emission ($P_{308}$) (Dusanter et al., 2008). Typical $R_{OH}$ values vary with ambient water-vapor concentrations due to collisional quenching of excited OH radicals and range from $1.5 - 4 \times 10^{-8}$ counts $s^{-1}$ /($cm^{-3}$ mW) (Fig. S3), with an estimated uncertainty of 18% (1$\sigma$) (Dusanter et al., 2008). Compared to the IU-FAGE instrument, the OH sensitivity of this detection cell is approximately a factor of 5-20 times lower per mW of laser power due to the multi-pass laser design of the IU-FAGE instrument compared to the single pass design described here. However, while the single pass design does not eliminate potential laser-generated interferences, it

significantly reduces laser-generated OH from reactions R4 and R5 as there is no beam overlap and the smaller beam size reduces the potential for double pulsing of the sampled air compared to the multi-pass design at the same laser power. This allows for higher laser powers to be employed in the single pass instrument, improving the limit of detection with significantly lower laser-generated interferences.

$$R_{OH} = \frac{S_{OH}}{[OH] \times P_{308}} \qquad (6)$$

A schematic of the calibrator is shown in Fig. S2. The calibrator consists of a rectangular flow reactor made of aluminum ($1.27 \times 1.27 \times 30$ cm) equipped with a quartz window on two sides (Dusanter et al., 2008). The light source is a low-pressure mercury lamp (UVP Inc) housed in an aluminum cartridge that is continuously purged with dry nitrogen to prevent light absorption by atmospheric gases as well as helping to stabilize the temperature of the lamp. A 10 SLPM flow of humidified air is used to create a turbulent flow in the reactor. Mixing ratios of water vapor and ozone are monitored in the flow exiting the calibrator using commercial analyzers (Dusanter

et al., 2008).

     Once a stable concentration of OH and $HO_2$ is produced in the calibrator after the lamp flux and water vapor concentration have stabilized, the photofragmentation efficiency (PE) of HONO is determined by adding an excess of NO (approximately 800 ppb) to the calibrator to convert the known concentrations of OH and $HO_2$ into HONO through the $HO_2 + NO \rightarrow OH + NO_2$ and $OH + NO \rightarrow$ HONO reactions. Figure 5 illustrates model simulations of the conversion of OH and $HO_2$ into HONO using the RACM2 mechanism

constrained to the concentrations of water vapor and oxygen. After production of OH and $HO_2$ in the illuminated region of the calibrator (first 10 ms), reactions with NO lead to the production of HONO after the approximate 80 ms residence time inside the calibrator. In these simulations, the photolysis of water vapor is adjusted to produce approximately 1 ppb of both OH and $HO_2$ in the calibrator, which in the absence of NO decreases after illumination due to loss from radical-radical reactions and surface loss (Fig. 5). During typical OH sensitivity calibrations, measurements of the loss of radicals in the absence of NO is measured by changing the location of the light

source relative to the exit of the calibrator (Dusanter et al., 2008). These measurements indicate that 20-30% of the OH and $HO_2$ radicals produced are lost due to reaction with the calibrator surfaces as well as loss due to the $OH + HO_2$ reaction. Model simulations indicate that a first order loss rate of 2.6 $s^{-1}$ is needed to match this observed loss of OH radicals in the calibrator in the absence of NO, and this

loss rate has been included in the simulations (Fig. 5). However, these simulations suggest that during photolysis efficiency calibrations, the excess of NO is sufficient to ensure that reaction with NO is the dominant radical sink accounting for greater than 95% of the total loss of OH, with less than 3% of the OH radicals lost via surface reactions and less than 2% lost by the OH + $HO_2$ and other radical-radical reactions. Model simulations of this chemistry also suggest that after addition of NO, the OH and $HO_2$ concentrations are negligible and the concentration of HONO is nearly equal to the total OH and $HO_2$ concentrations produced by the calibrator (Fig. 5).

Figure 6 illustrates a typical measurement of the photofragmentation efficiency. The original signal from the inital amount of OH produced in the calibrator in the absence of added NO is shown in panel (a) ($S_{initial\ OH}$), and the remaining OH concentration after NO is added to the calibrator is shown in panel (b) with the 355-nm photofragmentation laser blocked from entering the detection cell ($S_{remaining\ OH}$). This remaining OH signal is likely due to reactant segregation in the turbulent flow of the calibrator preventing all of the OH from reacting with the added NO. When the 355-nm photofragmentation laser is turned on, the increase in the signal relative to the remaining OH reflects the additional OH produced in the detection cell from HONO photolysis ($S_{HONO+remaining\ OH}$) (Fig. 6c). Model simulations indicate that reformation of HONO from reaction of the OH fragment with the added NO is negligible due to the reduced concentrations of both OH and NO in the low-pressure detection cell and the short reaction time between the photofragmentation and excitation laser pulses. Additionally, impurities in the NO cylinder appear to be photolyzed by the 355-nm photofragmentation laser leading to a small production of OH that is observed when the radical source in the calibrator is turned off (Fig. 6d). This impurity must be subtracted from the signal recovered as HONO and is measured by turning the mercury lamp off but keeping the 355-nm laser emission on ($S_{impurity}$). This impurity is not observed when the 355-nm laser is blocked from entering the detection cell. The source of this impurity is not clear but could be the result of heterogeneous reactions of $NO_2$ in the calibration system leading to the production of impurity HONO. Experiments employing the use of a scrubber to remove $NO_2$ from the cylinder such as iron(II) sulfate heptahydrate ($FeSO_4 \cdot 7H_2O$, Fisher scientific) did not appear to impact the OH signal due to the impurity, suggesting that production of this impurity may occur inside the NO cylinder. Additional experiments will be needed to identify this impurity.

The HONO photolysis efficiency (PE) of the 355-nm laser can be calculated as the ratio of OH signal recovered as HONO to the net OH and $HO_2$ concentrations produced in the calibrator. This can also be written as the ratio of net HONO signal to the initial OH signal, after corrections to account for the 20-30% OH radical loss due to the OH + $HO_2$ reaction and reaction on the walls of the calibrator based on measurements in the absence or NO as described above ($S_{initial\ OH,corr}$). Because the calibrator produces equal concentrations of OH and $HO_2$, the factor of 2 accounts for the conversion of the produced $HO_2$ to OH when NO is added.

$$PE = \frac{S_{HONO+remaining\ OH} - S_{remaining\ OH} - S_{impurity}}{2(S_{initial\ OH,corr,}) - S_{remaining\ OH}} \tag{7}$$

Photolysis efficiency measurements are typically performed before and after ambient measurement periods, and variations between measurement periods are likely caused by shifts in the alignment of the 355-nm photolysis laser. Typical PE values for the measurement periods described below were between 0.25% and 0.34% for a 355-nm laser power of approximately 1.5 W. Impurities in the added NO that react quickly with OH and compete with reaction of NO, such as $NO_2$, could lead to apparent lower photofragmentation efficiencies by reducing the amount of HONO produced in the calibrator. Model simulations suggest that a 5% $NO_2$ impurity could reduce the production of HONO by approximately 10% due to reaction of OH with $NO_2$ instead of NO (Fig, 5). As a result, the NO added should be of high purity, and chemical scrubbers designed to reduce impurities such as $NO_2$ should be used.

For ambient measurements, the concentration of HONO is determined from the net HONO signal ($Net_{HONO}$, Fig. 3) and the HONO sensitivity ($R_{HONO}$):

$$[HONO] = \frac{NET_{HONO}}{R_{HONO}} = \frac{NET_{HONO}}{R_{OH} \times PE} \tag{8}$$

For typical $R_{OH}$ values ranging from $1.5 – 4 \times 10^{-8}$ counts s$^{-1}$ /cm$^{-3}$ /mW and PE values between 0.25-0.34%, minimum detectable HONO concentrations (1$\sigma$) are typically between 9 and 18 ppt for a 10 min average in the absence of OH (laser power = 1.5 W at 355 nm, 1-3 mW at 308 nm). Because the limit of detection depends on the ambient OH signal that is subtracted from the HONO signal, the limit of detection reported here will be higher during the day compared to at night. For the highest sensitivity, 308 nm laser power, and photofragmentation efficiency described above, a daytime maximum concentration of OH of $4 \times 10^6$ cm$^3$ would increase the HONO limit of detection by approximately 20% (10 min average). The overall calibration uncertainty is estimated to be 35% (1$\sigma$), including the uncertainty associated with the OH calibration (18%, 1$\sigma$), and depends on the precision of the photofragmentation efficiency measurement. With the same parameters, the OH limit of detection is typically between 1.1 and $2.2\times10^6$ cm$^{-3}$ (S/N =1, 10 min average, laser power = 1-3 mW at 308 nm).

The limit of detection for HONO described above is similar to that reported by Dyson et al. (2021) in a laboratory setting using a similar instrument employing a 355 nm laser operating at 10 Hz for photofragmentation of HONO, and detection of OH at 308 nm using a dye laser operating at 5 kHz (12 ppt, 50-s average). While the details of the photofragmentation laser in this study were not provided, the lower repetition rate of the 355 nm laser likely leads to a higher pulse energy and a higher photofragmentation efficiency compared to the 10 kHz photofragmentation laser employed in this study. However, the higher pulse energy could lead to photolysis of other ambient species that produce OH and interfere with the measurements of HONO (see below). However, these potential interferences can be minimized in a laboratory setting.

As mentioned above, the large uncertainty associated with the photofragmentation measurements is likely due to shifts in the overlap between the two laser beams as a result of temperature fluctuations impacting the optical alignment. Although this uncertainty is currently large, measurements of HONO were in good agreement with an acetate CIMS instrument during the recent HOMEChem (House Observations of Microbial and Environmental Chemistry) indoor measurement campaign (Wang et al., 2020a). Overall, the measurements of HONO by the LP/LIF instrument agreed with the CIMS measurements to within ±20%, on average, suggesting that variations of the photofragmentation efficiency over the entire month-long campaign were less than the overall instrumental uncertainty reported here. An example of the measurements during the intercomparison is illustrated below, and a detailed analysis of the intercomparison, including an analysis of the spatial distribution of indoor HONO emissions, will be presented in a future publication.

## 3 Results and discussion

### 3.1 Outdoor measurements

The LP/LIF instrument was deployed in two locations on the Indiana University campus in Bloomington to measure outdoor concentrations of HONO as a test of the instrument. As HONO was the focus of these measurement periods, chemical modulation cycles were not performed, and the OH measurements presented may be impacted by unknown interferences. During the outdoor measurement periods described below the instrumental sensitivity to OH was ~$3 \times 10^{-8}$ counts s$^{-1}$ /(cm$^{-3}$ mW), and the measured photolysis efficiency was 0.25% for a laser power of approximately 1.5W at 355 nm, resulting in a limit of detection for HONO of approximately 18 ppt (10-minute average, 1$\sigma$), and a limit of detection for OH of approximately $5\times10^5$ cm$^{-3}$ (1-hour average, 1$\sigma$, 1-3 mW at 308 nm) (Table S1). The first of two measurement periods occurred during the summer of 2019 at a forested site within the Indiana University Research and Teaching Preserve (IURTP) (39.1908° N, 86.502° W), located approximately 2.5 km northeast of the IU campus (Lew et al., 2020). The mixed deciduous forest is dominated by emissions of isoprene and monoterpenes and is approximately 1 km away from the nearest high traffic road. The LP/LIF sampling axis was placed in a small clearing 5 meters from the IURTP field lab building, and sampling occurred approximately 0.5 meters above a grassy surface to measure potential HONO emissions from the soil. Additional measurements of NO

and $NO_2$ were conducted using a commercial chemiluminescence instrument (Thermo Environmental Instruments Inc. Model 42C) and measurements of the photolysis rate constant for $NO_2$ ($J_{NO_2}$) were conducted using a radiometer.

An average of HONO measurements from September 4–8, 2019 is presented in Fig. 7 (left). At this site, average mixing ratios of $NO_2$ varied from less than 500 ppt at night up to approximately 1 ppb during morning rush hour while mixing ratios of NO were below the detection limit of the instrument but estimated to be less than 300 ppt based on previous measurements of the $NO_2/NO$ ratio at this site (Lew et al., 2020). At night, HONO mixing ratios ranged from approximately 35 ppt to 75 ppt. Maximum observed HONO mixing ratios of approximately 150 ppt occurred during the day, indicating the presence of strong HONO sources that can compete with the loss of HONO due to photolysis during the daytime. Previous observations in remote or rural environments have indicated similar diel HONO trends, with average maximum mixing ratios on the order of 40 to 100 ppt (Huang et al., 2002; Acker et al., 2006; Zhou et al., 2011; Meusel et al., 2016). Similar to the results from these studies, the measured increase of the HONO mixing ratio during the day correlated with temperature and solar radiation, suggesting that the photolysis of $HNO_3$, biogenic release from soil, or other photo-enhanced HONO sources could all be relevant at the IURTP site. In addition, observed $HONO/NO_x$ values during the daytime are significantly higher than those reported from a majority of field campaigns (Elshorbany et al., 2012), but similar to observations from some rural, low-$NO_x$ environments (Zhou et al., 2007; Meusel et al., 2016), again highlighting the significance of local HONO sources other than heterogeneous $NO_x$ reactions.

Measurements of OH radical concentrations for these days by this instrument using spectral modulation to determine the instrument background reached a maximum value of $6\text{-}8 \times 10^6$ cm$^{-3}$, similar to that observed previously by the IU-FAGE instrument at this site after measured interferences were subtracted (Lew et al., 2020). While it is possible that the LP/LIF instrument is also be subject to unknown OH interferences, the single-pass laser system minimizes the effect of laser-generated interferences compared to the multipass system utilized by the IU-FAGE instrument. In addition, unknown interferences observed using chemical modulation by the IU-FAGE instrument correlated with increases in temperature. Average temperatures during the measurement periods described below were lower than the summer measurement period described in Lew et. al. (2020), suggesting that any unknown interferences may not have been detectable.

Nevertheless, because measurements of potential interferences using chemical modulation were not conducted during these days, the measured OH concentrations represent an upper limit to the actual OH concentrations, and the [HONO]/[OH] ratio a lower limit. The measured [HONO]/[OH] ratio was relatively constant during the day, resulting in an average value of approximately 1000 at this site. These values are greater than the estimated [HONO]/[OH] ratio assuming steady-state production and loss using reactions R2 and R1 and assuming the loss of HONO by reaction R3 is negligible compared to loss by photolysis (R1):

$$\frac{[HONO]}{[OH]}_{SS} \approx \frac{k_{OH+NO}[NO]}{J_{HONO}} \qquad (9)$$

Because the mixing ratio of NO was below the detection limit of the instrument (less than approximately 500 ppt) the concentration of NO used in this equation was estimated based on previous measurements of the $NO/NO_x$ ratio at this site. The value of J(HONO) was calculated as a function of solar zenith angle (Jenkin et al., 1997) and corrected for cloud coverage according to measured $J(NO_2)$ values. As illustrated in Fig. 7, the estimated [HONO]/[OH] ratio using this equation agrees with the measured ratio in the morning and evening but decreased in the afternoon to a value of approximately 10 as photolysis of HONO increased, returning to a value of approximately 1000 in the evening. The difference between the measured and modeled ratio reflects the magnitude of the missing HONO source in this environment, likely due to soil emissions mediated by the microbial community structure at this site (Mushinski et al., 2019).

A second outdoor measurement period occurred on the roof of the Multidisciplinary Science Building II near the center of the Indiana University Bloomington campus. Sampling occurred at a height of approximately 12 m above the ground, and 1-5 m away from building surfaces. This measurement site is adjacent to local roads and is influenced by direct emissions from traffic and other

anthropogenic sources. Average measurements from October 14th and 15th, 2019 are shown in Fig. 7 (right). At this location, mixing ratios of NO were typically 0.25 ppb at night and increasing to approximately 0.7-1 ppb during the morning and evening rush hours. $NO_2$ varied from 0.5 to 2 ppb during the day and night, higher that that observed at the IURTP site. Overall, HONO mixing ratios were higher than those observed at the IURTP, with maximum mixing ratios of 350-400 ppt at night decreasing to 50-100 ppt during the day.

Similar to previous observations in urban environments, maximum HONO values were observed at night and during morning traffic and correlated with $NO_x$, indicating that direct emissions and $NO_2$ conversion on buildings and other urban surfaces were more relevant in this environment (Michoud et al., 2014; Czader et al., 2015; Xu et al., 2015). The observed ratio of HONO to $NO_x$ of approximately 10-30% was higher than the 1-8% typically observed in other in other urban environments (Elshorbany et al., 2012), especially at night, which may reflect the close proximity of the LP/LIF inlet to nearby building surfaces and potential surface-driven HONO sources.

Measured OH concentrations by the instrument were similar to that observed at the forested site, with maximum observed concentrations of 6-8 $\times 10^6$ $cm^{-3}$. As at the forest site, chemical modulation experiments to measure unknown interferences were not done during these days, thus these OH measurements represent an upper limit to the actual OH measurements. In contrast to the forest site, the measured [HONO]/[OH] ratio at the urban site varied during the course of the day, decreasing from a value of approximately $10^4$ during the morning to a value of approximately 100 during the afternoon, and increasing to a value of $10^4$ in the evening. This trend is similar to

that estimated by the steady-state [HONO]/[OH] ratio using equation 9. While the ratio estimated by this equation reproduces the measured ratio in the afternoon and evening, it underestimates the measured ratio in the morning, again suggesting that there is an additional source of HONO at this site either from direct emissions or heterogeneous production.

**3.2 Indoor measurements**

In 2018, the LP/LIF instrument was deployed inside a test house in Austin, Texas as part of the HOMEChem study (Farmer et al., 2019).

HOMEChem was a collaborative field study intended to investigate how household activities influence emissions and chemistry of gases and particles within the indoor environment. During the campaign HONO was measured by the Indiana University LP/LIF instrument as well a high-resolution time-of-flight chemical ionization mass spectrometer (HR-ToF-CIMS) from the University of Toronto (Collins et al., 2018). Several cooking and cleaning perturbation experiments were performed from June 5-28. During this period the instrumental sensitivity to OH was ~2.75 $\times 10^{-8}$ counts $s^{-1}$ /($cm^{-3}$ mW), and the measured photolysis efficiency was 0.34%

(1.5W at 355 nm), resulting in a limit of detection for HONO of approximately 9 ppt (10 min average, 1$\sigma$, 1-3 mW at 308 nm) (Table S1). Results from these experiments are summarized in Wang et al. (2020a).

The LP/LIF sampling axis was placed in the living area of a 111 $m^2$ manufactured home, adjacent to two westward-facing windows. Results from a repeated set of enhanced ventilation experiments, where all of the windows and doors of the house were opened, are presented in Fig. 8. During the ventilation periods the HONO mixing ratio was reduced to approximately 1 ppb due to

405 mixing with outdoor air and returned to a steady state of 3-4 ppb within 20-30 minutes after ventilation was stopped. These values are similar to those measured by the University of Toronto CIMS instrument during this experiment (Wang et al., 2020b), and a more complete instrumental intercomparison of the HOMEChem HONO measurements will be presented in a future publication. The fast return to steady-state concentrations after ventilation ceased indicates that gas phase HONO is in equilibrium with a reservoir of HONO precursors on interior surfaces. Similar results from a set of residential ventilation experiments in a previous study further suggests that

indoor HONO mixing ratios are governed by dynamic partitioning with a surface reservoir (Collins et al., 2018). Additionally, during this experiment, the air conditioning unit in the house was turned off to minimize variations due to the on-off cycling of the system. As a result, the indoor temperature of the house slowly increased until the unit was turned on at the end of the experiment. The observed increase in HONO as the temperatures increased within the house during the experiment suggests that the equilibrium is temperature dependent (Fig. 8).

Figure 9 illustrates an example of the LP/LIF measurements and the University of Toronto CIMS instrument (Collins et al., 2018) during a cooking event using a gas stove during HOMEChem. During this experiment, mixing ratios of HONO were approximately 2 ppb for several hours prior to the cooking episode. When the gas stove was turned on, mixing ratios of HONO quickly increased to approximately 6 ppb before slowly decaying after the gas stove was turned off. As illustrated in this figure, the LP/LIF measurements of HONO were in excellent agreement with the CIMS measurements during this event, with the measurements agreeing

to less than 20%, and are representative of the overall agreement during the intercomparison, which will be presented in a future publication. These results provide confidence in the accuracy of the LP/LIF instrument and the calibration method.

**3.3 Potential interferences**

Potential interferences with outdoor measurements of HONO include species that photolyze at 355-nm leading to both prompt and

secondary production of OH in the detection cell. Possible prompt interferences include $HNO_3$, $H_2O_2$, and other organic peroxides, while potential secondary interferences include species that could produce OH precursors from photolysis, such as $HO_2$ from the photolysis of HCHO and $HO_2NO_2$, which could react to produce OH. A previous analysis of the impact of these species on the atmospheric pressure LP/LIF instrument by Liao et al. (2006a) suggested that the photolysis of typical ambient mixing ratios of these species would not lead to the production of significant OH concentrations at a photofragmentation laser pulse energy of approximately 700 mJ at 355-nm, with

1 ppb of HCHO and 1ppb $HO_2NO_2$ together estimated to produce 0.16 ppt of OH, 1ppb of $H_2O_2$ estimated to produce 1.1 ppt of OH, and 1ppb of $HNO_3$ estimated to produce 0.03 ppt of OH (Liao et al., 2006a). Given that the LP/LIF instrument described here utilizes a much lower pulse energy (0.2 mJ) and the mixing ratios of potential interfering species are reduced upon sampling at low pressure, it is likely that any OH produced by photolysis of these species would be significantly less than that estimated by Liao et al. (2006a) and would not significantly interfere with outdoor HONO measurements. In addition, the short time interval between the 355 and 308-nm

laser pulses likely minimizes these and other secondary interferences.

During HOMEChem measurements of gas phase organics were generally higher indoors than outdoors across a broad range of species that were further enhanced during cooking events (Farmer et al., 2019). The cooking events likely increased the concentration of formaldehyde, as combustion from gas stoves can be a significant source of formaldehyde, resulting in indoor mixing ratios potentially greater than 10 ppb (Salthammer et al., 2010; Logue et al., 2014). The agreement of the LP/LIF instrument with the CIMS instrument

during these events such as that illustrated in Figure 9 suggests that interferences from the photolysis of formaldehyde or the potential reduction of the photofragmentation efficiency due to absorption of the 355 nm laser by formaldehyde are likely minimal. Unfortunately, formaldehyde concentrations were not quantified during HOMEChem, and additional measurements together with quantification of formaldehyde concentrations will be needed to confirm these results.

However, higher concentrations of other species that photolyze at 355-nm leading to the formation of OH could interfere with

measurements of HONO. One possible indoor interference is HOCl, which can be produced during chlorine bleach mopping episodes. Figure 9 also shows a bleach mopping experiment during HOMEChem that resulted in the production of approximately 100-200 ppb of HOCl. During several of these episodes, measurements of HONO by the LP/LIF increased and were correlated with the HOCl measurements while the HONO measurements by the University of Toronto CIMS instrument decreased as expected during these bleach mopping episodes, as the increase in pH likely impacted the surface equilibrium production of HONO (Collins et al., 2018; Wang et al.,

2020b). Although the absorption cross section of HOCl is approximately a factor of 40-50 times lower than that of HONO ($1.2 \times 10^{-20}$ $cm^2$) (Burkholder et al., 2019) the indoor mixing ratios of HOCl during several of these mopping episodes were 50-100 times greater than the mixing ratios of HONO during these experiments. Bleach mopping experiments that produced lower mixing ratios of HOCl (less than 12 ppb) resulted in lower interferences in the LP/LIF instrument and better agreement with the CIMS instrument. This

interference will be examined in more detail in a future publication. It appears unlikely that outdoor mixing ratios of HOCl would be a significant interference given that HOCl mixing ratios in marine environments are generally less than 1 ppb (Lawler et al., 2011), much lower than the level of indoor HOCl that produced an interference during HOMEChem. While the absorption cross sections of HOBr and HOI are factors of approximately 10 and 30 times greater than that of HOCl at 355 nm (Burkholder et al., 2019), outdoor ambient mixing ratios of HOBr and HOI are much lower than ambient HOCl, with measured values of HOBr less than 26 ppt in the arctic marine boundary layer (Liao et al., 2012), and measured mixing ratios of HOI less than 70 ppt in the marine boundary layer (Tham et al., 2021). Thus, it is unlikely that outdoor mixing ratios of HOBr and HOI would significantly interfere with LP/LIF measurements of HONO. However, these and other potential interferences, both prompt and secondary, will need to be tested in the laboratory.

Because the HONO measurements require the subtraction of the fluorescence signal due to ambient OH, the limit of detection of the instrument will vary with the concentrations of ambient OH as discussed above. The worst limit of detection will likely occur around solar noon when elevated photolysis frequencies lead to large production rates for OH and a short lifetime for HONO. In addition, while the current single-pass laser system with excitation at 308 nm using a high repetition rate laser reduces laser-generated OH from the photolysis of ozone, unknown interferences leading to the formation of OH radicals inside the detection cell, such as Criegee intermediates formed from the ozonolysis of alkenes (Rickly and Stevens, 2018) could interfere with measurements of OH in addition to impacting the HONO limit of detection. These and other unknown interferences can be measured through the external removal of ambient OH through the chemical modulation technique described above, similar to that used by the IU-FAGE instrument (Griffith et al., 2016; Rickly and Stevens, 2018; Lew et al., 2020). In the absence of an interference, this method could also be used to improve the limit of detection of HONO through the removal of the ambient OH background signal.

## 4 Conclusions

The LP/LIF instrument described here demonstrates a sensitivity and limit of detection for HONO that is sufficient for ambient measurements of HONO in both indoor and outdoor environments. Incorporating two separate lasers and employing excitation and detection of the OH fragment at 308 nm in addition to ambient sampling at low pressure minimizes interferences from laser generated OH that may have impacted previous LP/LIF measurements of HONO at atmospheric pressure (Liao et al., 2006a). The LP/LIF instrument has a 1σ detection limit for HONO of approximately 9 ppt for a 10-min integration time using 1.5W of radiation at 355 nm and a repetition rate of 10 kHz for photofragmentation of HONO, and 1-3 mW at 308 nm and a repetition rate of 10 kHz for detection of the OH fragment. The instrument is calibrated by determining the photofragmentation efficiency of HONO and sensitivity to detection of the OH fragment through the titration of a known concentration of OH from the photo-dissociation of water vapor with nitric oxide to produce a known concentration of HONO. Measurement of the concentration of the OH radical fragment relative to the concentration of HONO provides a measurement of the photofragmentation efficiency. The overall calibration uncertainty is estimated to be 35% (1σ), including the uncertainty associated with the OH calibration (18%, 1σ), and depends on the precision of the photofragmentation efficiency measurement. The current limit of detection of HONO can be improved by increasing the photofragmentation efficiency with a more powerful laser system, increasing the OH detection efficiency by increasing the laser power at 308 nm, and through improvements to fluorescence detection efficiency and the overlap of the photofragmentation and excitation lasers.

The LP/LIF instrument has several advantages compared to other instrumental techniques, as the technique is free of inlet artifacts such as heterogeneous formation or loss of HONO on surfaces and likely has minimal interferences from other atmospheric species. The ability to conduct near simultaneous measurements of both HONO and OH concentrations by this instrument will allow more accurate measurements of the [HONO]/[OH] ratio than individual measurements of each by separate instruments, given that the

uncertainty associated with the OH detection sensitivity cancels out in measurements of the ratio in the LP/LIF instrument. Measurements of the [HONO]/[OH] ratio in comparison to model predictions can provide important information concerning the relative contribution of sources other than gas-phase production and loss by reactions R1-R3, as illustrated in Fig. 7 and discussed above. Furthermore, the calibration method involving production of HONO from the OH + NO reaction using the established calibration method for generating a known amount of OH provides a simple and robust calibration for both OH and HONO in field settings.

In addition to improving the sensitivity of the instrument, future work will involve identification and quantification of some of the potential interferences in the measurements of HONO and OH by the LP/LIF instrument as discussed above as well as improving the stability of the beam overlap in order to improve the precision associated with the photofragmentation efficiency calibration method. In addition, the measured HONO sensitivity by the photofragmentation calibration method will be compared to that determined through the production of HONO by the reaction of gas-phase hydrochloric acid in a humidified gas stream with solid sodium nitrite (Febo et al., 1995; Gingerysty and Osthoff, 2020; Lao et al., 2020). While this calibration method is not as simple to implement in the field in addition to requiring quantification of the HONO produced, comparison of the instrument sensitivity derived from this calibration source in the laboratory would provide additional confidence in the calibration of the instrument by the photofragmentation method.

*Data availability.* Data are available upon request from the corresponding author (pstevens@indiana.edu).

*Author contributions:* SD, LM, BB, ER and PS contributed to the design and construction of the instrument. BB, ER, and PS were responsible for the calibrations and measurements. BB, ER and PS wrote the manuscript with contributions from all coauthors.

*Competing interests.* The authors declare that they have no conflict of interest.

*Acknowledgements.* We would like to thank James Flynn (University of Houston) for the spectroradiometer used to obtain the $J(NO_2)$ measurements. The CIMS measurements of HONO are courtesy of Prof. Jon Abbatt (University of Toronto), and the HOCl measurements are courtesy of Prof. Delphine Farmer (Colorado State University).

*Financial support.* This study was supported by the National Science Foundation, Directorate for Geosciences (grant nos. AGS-1012161 and AGS-1827450) and the Alfred P. Sloan Foundation, Chemistry of Indoor Environments Program (grant nos. G-2017-9944 and G-2018-11061).

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

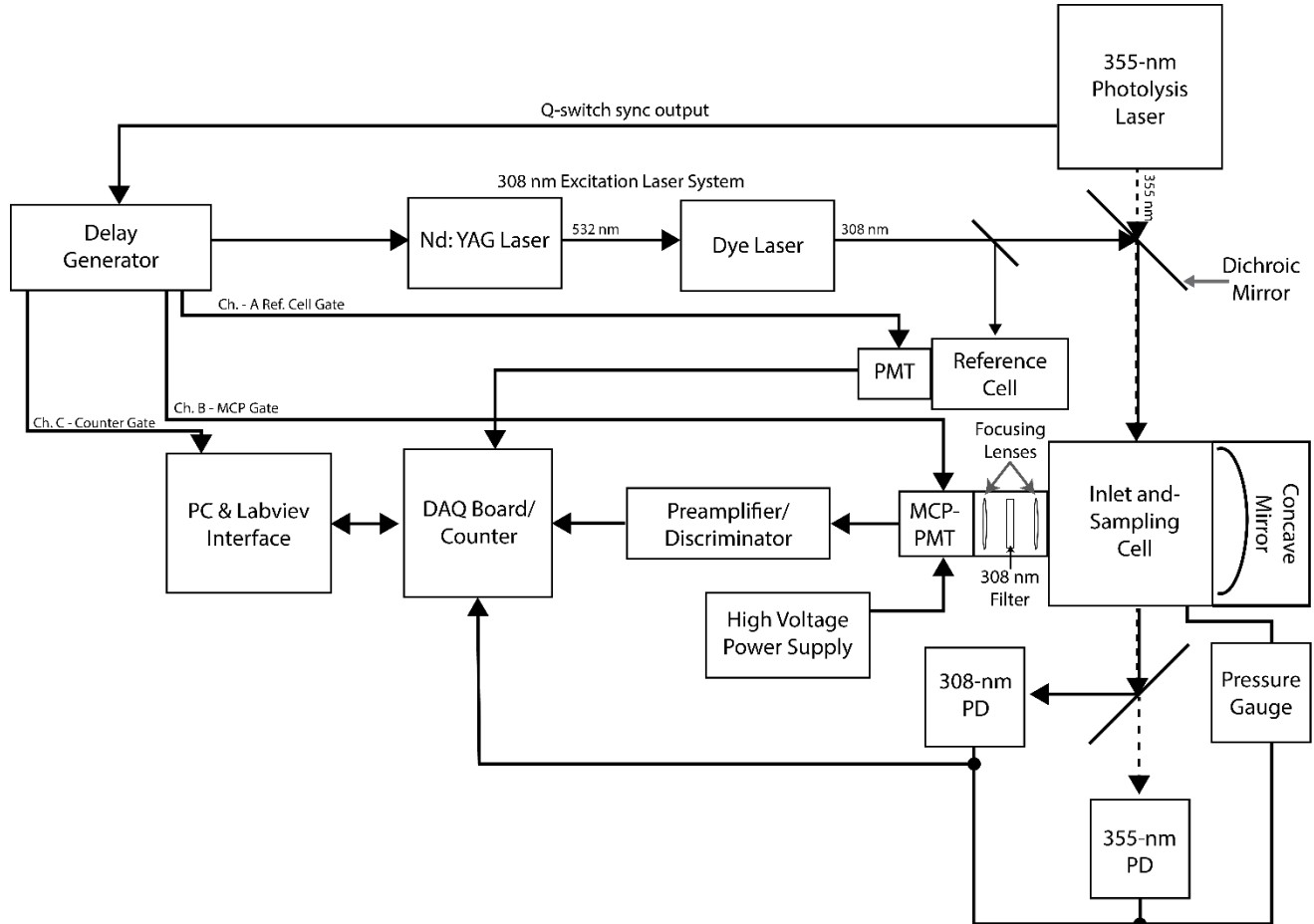

**Figure 1:** Schematic diagram of the LP/LIF instrument.

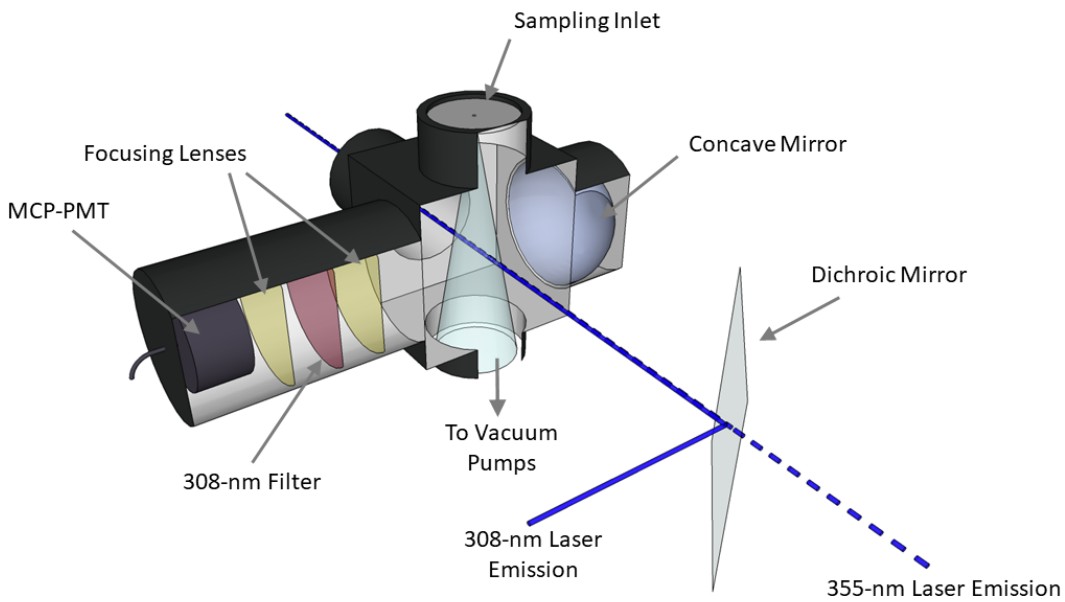

**Figure 2:** Diagram of the LP/LIF sampling cell.

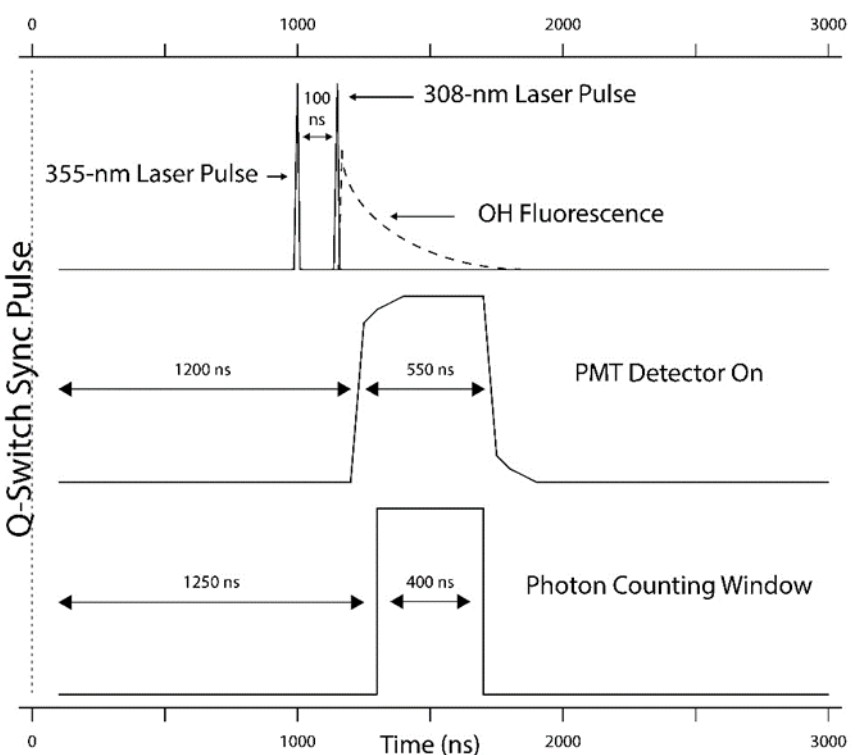

**Figure 3:** Timing schematic depicting one photofragmentation/excitation/detection cycle.

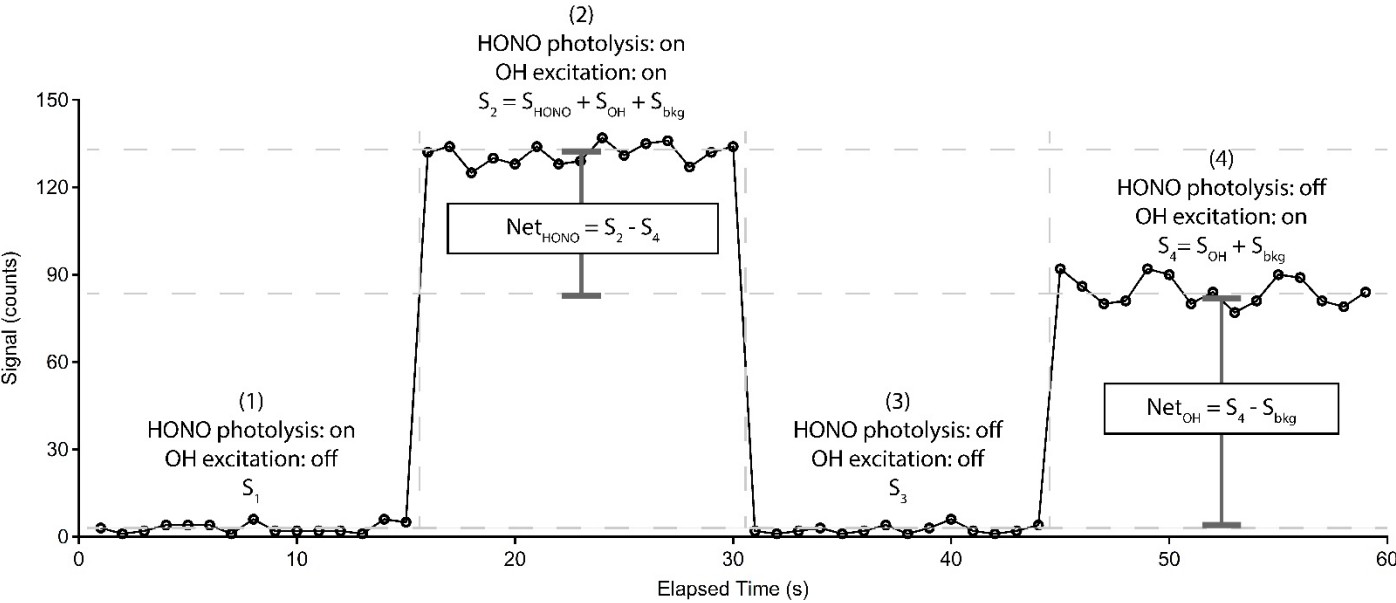

**Figure 4:** Sample measurement cycle from the LP/LIF instrument during measurement of OH and HONO during laboratory calibrations. The raw signal has not been normalized for the power of the excitation laser or the total radical concentration produced by the calibrator. $S_{bkg}$ is the average of the offline signals $S_1$ and $S_3$

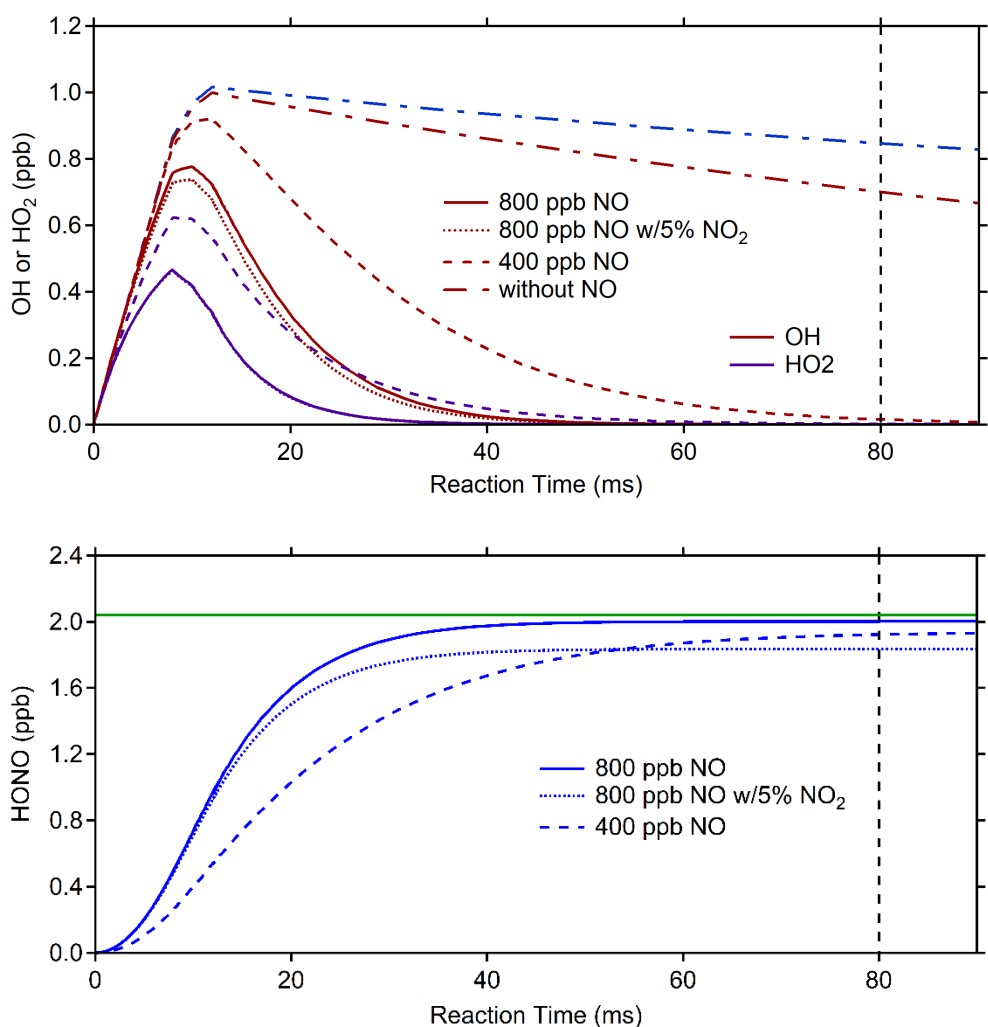

**Figure 5:** Photolysis efficiency (PE) calibration modeling. The top plot illustrates the modeled evolution of OH (red lines) or $HO_2$ (blue lines) in the presence of 400 (dashed line) or 800 ppb (solid line) of NO, while the bottom plot shows the modeled HONO production (blue lines). The vertical black dashed line represents the approximate reaction time between the onset of radical production from 184.9-nm photolysis of water and the exit of the calibration source at a flow rate of 10 SLPM. The horizontal green line represents the total radicals produced in the photolysis region in the absence of NO. The dotted line represents a simulation with 800 ppb of NO with a 5% $NO_2$ impurity (see text).

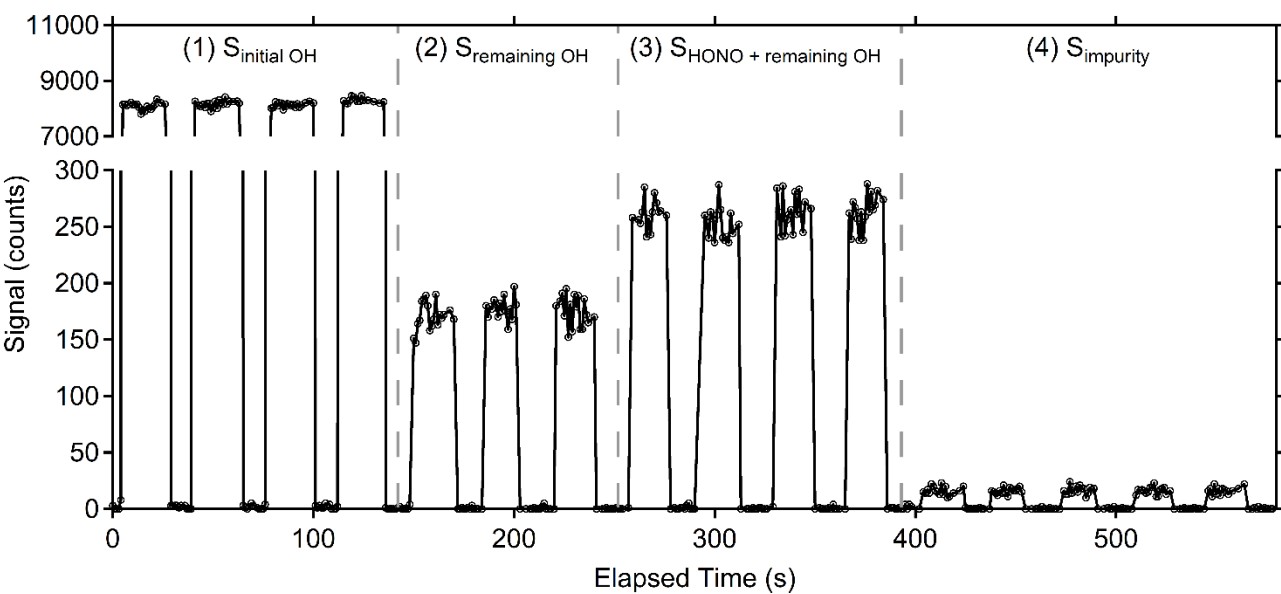

**Figure 6:** Example of photofragmentation efficiency measurements from an OH + NO → HONO calibration. (**a**) The signal observed from total amount of OH produced in the calibrator ($S_{initial\ OH}$). (**b**) Signal observed from remaining OH after the addition of NO converts the majority of radicals to HONO ($S_{remaining\ OH}$). (**c**) Sum of signal from remaining OH and signal from OH produced in the detection cell after HONO photolysis ($S_{HONO+remaining\ OH}$). (**d**) Signal from the 355-nm photolysis of impurities in the NO cylinder observed when the radical source in the calibrator is turned off ($S_{impurity}$).

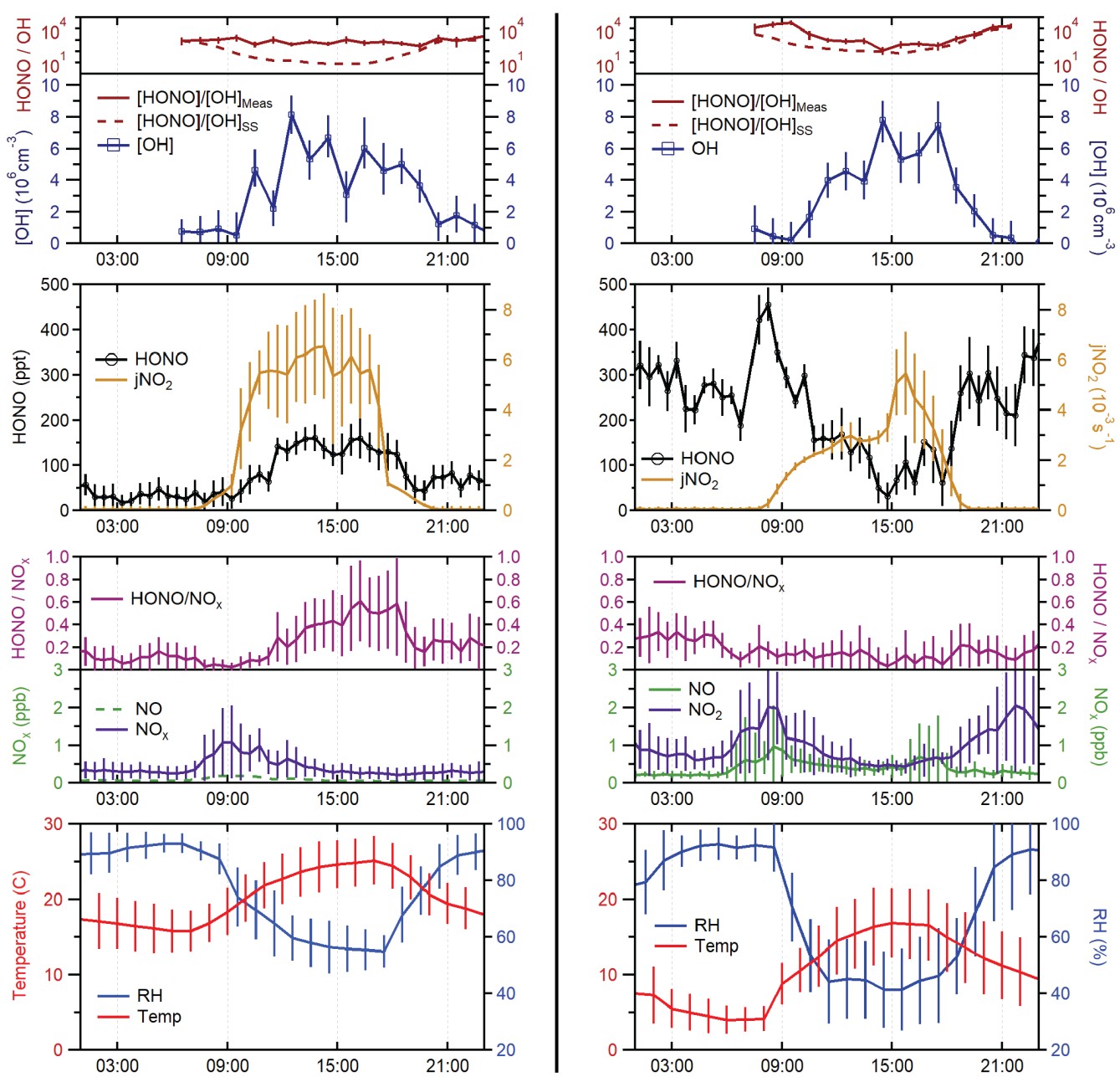

**Figure 7:** Average measurements of HONO, [OH], HONO/OH, J(NO₂), temperature, RH, NOx, and HONO/NOₓ from the forested site (left) and urban site (right). Mixing ratios of NO at the forested site were estimated based on previous measurements of the NO/NO₂ ratio at this site (dashed green line). During the measurement period conducted at the urban site, the detection cell was partially shaded by the building during the morning and early afternoon, resulting in the observed peak in J(NO₂) at 15:00. Error bars represent the standard deviation of the diurnal average measurements (1σ).

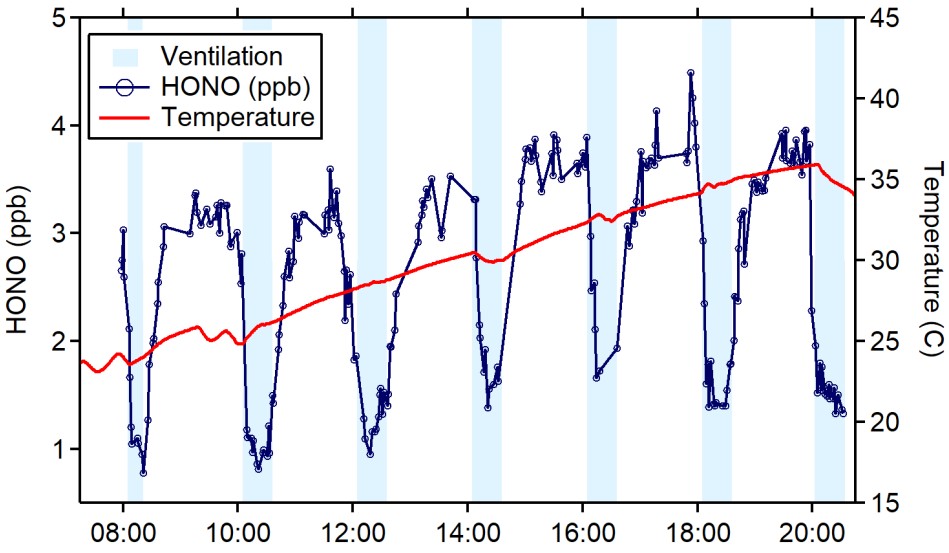

**Figure 8:** HONO (blue) and temperature (red) data from the June 4th ventilation experiment during the HOMEChem study. Shaded areas represent ventilation periods (open doors and windows).

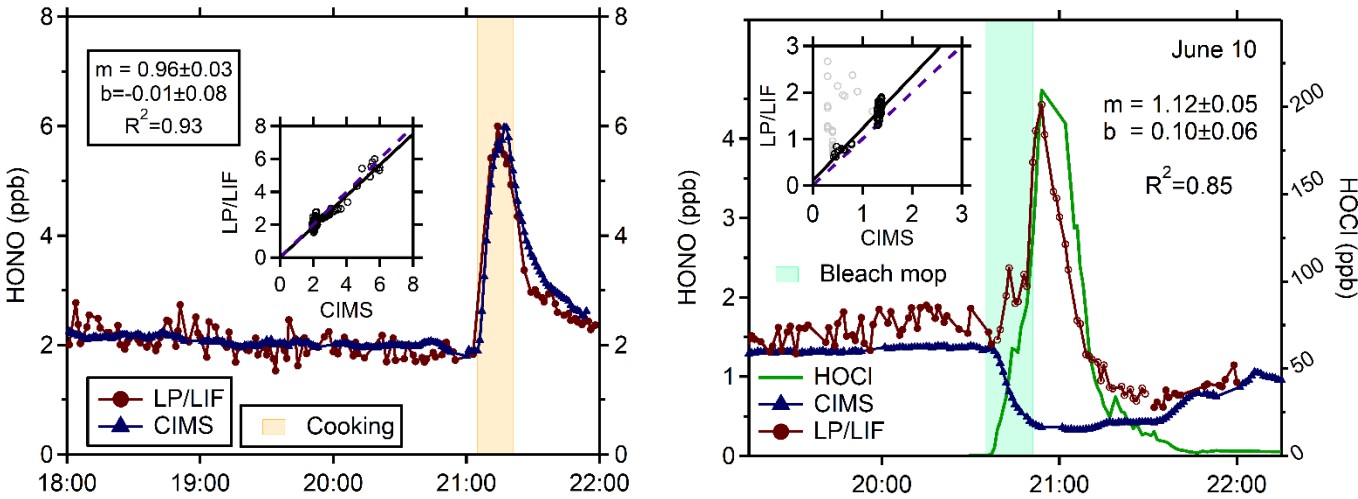

**Figure 9:** Measurements of HONO by the LP/LIF instrument (red points) and the University of Toronto CIMS instrument (blue points) from a cooking experiment (left) and a bleach mopping experiment (right) during the HOMEChem study illustrating the interference in the LP/LIF instrument from HOCl. The correlation coefficients for the bleach mopping experiment exclude the LP/LIF measurements when HOCl was elevated.