# Peer review of "Figure S1: Absorption cross section of HONO with the 355-nm emission from the third harmonic of the Nd:YLF laser highlighted."

_Atmospheric Measurement Techniques, 2021_

## Author Comment (AC1)

We would like to thank the reviewers for their efforts in reviewing this manuscript, and we feel that the manuscript is much stronger with their suggested changes. Below are detailed responses to their comments, which are highlighted in italics.

Reviewer #1

The authors aim for describing a new instrument for the detection of HONO by laser-photofragmentation of HONO and subsequent detection of OH by LIF. Although the topic of the paper would have been in the scope of this journal, the way the authors structure the paper is not appropriate. The authors give an adequate clear description of the instrument and its calibration.

However, the results and discussion part is not fitting the scope of the journal. The authors mainly describe measurements and try to give a chemical explanation of the observed concentrations. This is clearly out of the scope of this journal and should be significantly shortened. Unfortunately, the authors give only little results and discussion of the performance of the instrument. Even the discussion of interferences is kept on a level of estimates from literature and the only experimental result is only mentioned to be discussed in future publications. More detailed experimental investigations in the laboratory could have been done.

A comparison of the measured HONO/OH ratio with calculations could have been valuable, if this is used to test the validity of the instrument. However, this is likely not possible, because concentrations are not only determined by the photo-stationary state of gas-phase reactions and the additional measurements may be incomplete or not of high-enough quality. Another option would have been to include a detailed comparison with the CIMS instrument, but the author decided not to do so, but to shift this to another publication.

Overall, the authors mention topics that could have been discussed to show and discuss the performance of the instruments even without further experiments, which would have been even better. Because of the lack of content, I recommend rejecting the paper. It might become suitable for the journal, if there is a broader discussion of results.

*As suggested, we have included a few examples of the intercomparison with the University of Toronto CIMS instrument during the HOMEChem campaign, illustrating the excellent agreement between the instruments, and we have expanded the discussion of potential interferences. We have included two examples of the intercomparison in Figure 9 of the revised manuscript. The first example illustrates the agreement between the instruments during a cooking episode, and is discussed on pages 11-12 of the revised manuscript:*

[revised manuscript text omitted]

*We feel that the inclusion of examples of the instrument intercomparision, as well as the expanded discussion of interferences as suggested, better illustrates the performance of the LP/LIF instrument.*

Detailed comments to the part of the experimental section:

L13: Full calibration of the sensitivity requires also the calibration of the OH detection sensitivity.

*We have clarified that the HONO calibration also requires calibration of the OH detection sensitivity in the abstract of the revised manuscript as suggested.*

The LP/LIF instrument is calibrated by determining the photo-fragmentation efficiency of HONO and calibrating the instrument sensitivity for detection of the OH fragment.

L16: Does the LOD refer to HONO and OH?

*We have clarified that the LOD refers to the HONO limit of detection in the abstract.*

The LP/LIF instrument has demonstrated a $1\sigma$ detection limit for HONO of 9 ppt for a 10-min integration time.

L39: The explanation of HONO accumulating during nighttime is a bit misleading, because gas-phase reaction alone would not explain the increase of HONO during the night, but only shift the photo-stationary state to HONO at dawn.

*We agree and have removed this statement from the revised manuscript.*

L166: Here or somewhere else the authors should mention the duration of each step in the measurement cycle.

*We have clarified that the typical duration in each step is 15-s on page 5 of the revised manuscript as suggested.*

Each measurement cycle consists of four 15-s steps – (1) a background signal is established where HONO is photolyzed but the 308-nm laser is tuned off-resonance ($S_1$), (2) both ambient OH and the OH fragment from HONO are excited by tuning the 308 nm laser to on resonance ($S_2$), (3) the 355-nm photolysis laser is blocked by a shutter and background signal is re-established by tuning the 308-nm laser off-resonance ($S_3$), and (4) the 355-nm laser is still blocked but ambient OH is excited by tuning the 308-nm laser on-resonance ($S_4$).

L211/L213: There is inconsistency in the naming of the quantum yield.

*This typo has been corrected.*

L233/234: The term "effective sensitivity" is rather confusing in this context. The sensitivity of the instrument does not depend on laser power due to the normalization of the fluorescence signal to the laser power. The authors likely mean a better limit of detection that can be achieved at higher laser power, because the total fluorescence counts increase and is therefore more likely larger than the noise. Please clarify.

*We have replaced the term "effective sensitivity" with "limit of detection" on page 7 of the revised manuscript as suggested.*

L233/234: The authors argue that photolytic interferences become smaller in the single-pass configuration compared to the multi-pass configuration. This would allow to operate the system at higher laser power. This is rather confusing because at higher laser power photolytic interferences will again gain in importance. Please clarify.

*While the single pass design does not eliminate potential laser generated interferences, we have clarified that the single pass design significantly reduces laser-generated interferences compared to that produced by the multi-pass design at the same laser power. This has been clarified on page 7 of the revised manuscript.*

However, while the single pass design does not eliminate potential laser-generated interferences, it significantly reduces laser-generated OH from reactions R4 and R5 as there is no beam overlap and the smaller beam size reduces the potential for double pulsing of the sampled air compared to the multi-pass design at the same laser power. This allows for higher laser powers to be employed in the single pass instrument, improving the limit of detection with significantly lower laser-generated interferences.

L242: What do the authors mean with "once a stable concentration OH and HO2 is produced"? What does need to stabilize?

*The concentration of OH and $HO_2$ depend on both the lamp flux and water vapor concentration, which take some time to stabilize at the start of the calibration. This has been clarified on page 7 of the revised manuscript.*

Once a stable concentration of OH and $HO_2$ is produced in the calibrator after the lamp flux and water vapor concentration have stabilized, the photofragmentation efficiency (PE) of HONO is determined by adding an excess of NO (approximately 800 ppb) to the calibrator to convert the known concentrations of OH and HO2 into HONO through the $HO_2 + NO \rightarrow OH + NO_2$ and $OH + NO \rightarrow HONO$ reactions.

L249: How was the loss of 5% determined?

*The 5% loss was determined by model simulations of the chemistry. This has been clarified on page 7 of the revised manuscript as described below in response to the following comment.*

L250: It would be beneficial for the reader to know the wall loss rates that are assumed and to specify the fractional loss to specific loss processes.

*The loss of radicals in the calibrator in the absence of NO was measured as described previously by changing the location of the light source in the calibrator (Dusanter et al., 2008). The measured decrease in the concentration of OH as a function of distance from the exit of the calibrator was found to be approximately 20-30% for a reaction time of 80 ms and a flow rate of 10 slpm. In the absence of added NO this loss rate is due to both loss on walls of the calibrator as well as loss due to radical-radical reactions such as the $OH + HO_2$ reaction. Model simulations were then conducted to determine the relative contribution of radical-radical reactions and wall loss to the overall loss of radicals in the calibrator. From these simulations, a first order loss rate of 2.6 $s^{-1}$ is needed to match the observed loss of OH radicals in the calibrator in the absence of NO. When 800 ppb of NO is added to the calibrator, model simulations suggest that wall loss in the calibrator contributes to less than 3% of the total loss of OH radicals, with the $OH + HO_2$ reaction contributing to less than 2%, as the $OH + NO$ reaction contributes to greater than 95% of the total loss of OH radicals. This has been clarified on pages 7-8 of the revised manuscript, and we have included examples of these simulations as a new Figure 5 in the revised manuscript.*

Figure 5 illustrates model simulations of the conversion of OH and $HO_2$ into HONO using the RACM2 mechanism constrained to the concentrations of water vapor and oxygen. After production of OH and $HO_2$ in the illuminated region of the calibrator (first 10 ms), reactions with NO lead to the production of HONO after the approximate 80 ms residence time inside the calibrator. In these simulations, the photolysis of water vapor is adjusted to produce approximately 1 ppb of both OH and $HO_2$ in the calibrator, which in the absence of NO decreases after illumination due to loss from radical-radical reactions and surface loss (Fig. 5). During typical OH sensitivity calibrations, measurements of the loss of radicals in the absence of NO is measured by changing the location of the light source relative to the exit of the calibrator (Dusanter et al., 2008). These measurements indicate that 20-30% of the OH and $HO_2$ radicals produced are lost due to reaction with the calibrator surfaces as well as loss due to the OH + $HO_2$ reaction. Model simulations indicate that a first order loss rate of 2.6 $s^{-1}$ is needed to match this observed loss of OH radicals in the calibrator in the absence of NO, and this loss rate has been included in the simulations (Fig. 5). However, these simulations suggest that during photolysis efficiency calibrations, the excess of NO is sufficient to ensure that reaction with NO is the dominant radical sink accounting for greater than 95% of the total loss of OH, with less than 3% of the OH radicals lost via surface reactions and less than 2% lost by the OH + $HO_2$ and other radical-radical reactions.

L252: The text sounds as if there is a significant fraction of OH left, but Fig S4 suggests that this is negligible.

*We have revised this text as suggested on page 8 of the revised manuscript.*

Model simulations of this chemistry also suggest that after addition of NO, the OH and $HO_2$ concentrations are negligible and the concentration of HONO is nearly equal to the total OH and $HO_2$ concentrations produced by the calibrator (Fig. 5).

L252: Can the authors exclude that reactions of NO from the calibration source leads to any back-reaction of OH to HONO after the 355nm laser pulse has been applied in the measurement cell?

*Model simulations indicate that reformation of HONO from reaction of the OH fragment with the added NO is negligible due to the reduced concentrations of both OH and NO in the low-pressure detection cell and the short reaction time between the photofragmentation and excitation laser pulses. This has been clarified on page 8 of the revised manuscript.*

Model simulations indicate that reformation of HONO from reaction of the OH fragment with the added NO is negligible due to the reduced concentrations of both OH and NO in the low-pressure detection cell and the short reaction time between the photofragmentation and excitation laser pulses.

Figure 4/5: Are really counts shown or normalized count rates? Why are numbers in Fig. 4 so much smaller compared to numbers in Fig. 5, if they are also derived from calibration measurements?

*Both figures illustrate the measured count rates and are not normalized for differences in the excitation laser power or the initial radical concentration between the two different calibration experiments, which accounts for the factor of 2 difference in the measured counts illustrated in these figures. This has been clarified in the caption of Figure 4.*

Figure 4: Sample measurement cycle from the LP/LIF instrument during measurement of OH and HONO during laboratory calibrations. The raw signal has not been normalized for the power of the excitation laser or the total radical concentration produced by the calibrator. $S_{bkg}$ is the average of the offline signals $S_1$ and $S_3$

L267: It is not very clear for the reader, which correction is applied to S_OH. Is this needed because different losses apply, if NO is added or not? Number of corrections may help to better understand what is done.

*This is the correction to account for the 20-30% loss of radicals in the absence of NO due to both radical-radical reactions and radical loss of the walls of the calibrator measured as described above. This has been clarified on page 8 of the revised manuscript.*

This can also be written as the ratio of net HONO signal to the initial OH signal, after corrections to account for the 20-30% OH radical loss due to the OH + $HO_2$ reaction and reaction on the walls of the calibrator based on measurements in the absence or NO as described above ($S_{initial\ OH,corr}$).

L271: The authors mention several possible problems with impurities of the NO added in the calibration procedure. This discussion should be extended by a quantitative estimate, if these reactions could play a role for the conditions described in this work.

*We have included model simulations to illustrate the potential impact of NO2 impurities in the added NO on page 8 of the revised manuscript, as suggested.*

Impurities in the added NO that react quickly with OH and compete with reaction of NO, such as $NO_2$, could lead to apparent lower photofragmentation efficiencies by reducing the amount of HONO produced in the calibrator. Model simulations suggest that a 5% $NO_2$ impurity could reduce the production of HONO by approximately 10% due to reaction of OH with $NO_2$ instead of NO (Fig, 5). As a result, the NO added should be of high purity, and chemical scrubbers designed to reduce impurities such as $NO_2$ should be used.

L282: It would help to give a quantitative estimate about the impact of a typical additional OH concentration during midday on the limit of detection of HONO.

*Using the maximum OH sensitivity, laser power, and photofragmentation efficiency described on page 8 of the revised manuscript, a daytime maximum concentration of OH of approximately $4 \times 10^6$ $cm^{-3}$ would increase the estimated HONO limit of detection by approximately 20%. This has been clarified on page 9 of the revised manuscript.*

For the highest sensitivity, 308 nm laser power and photofragmentation efficiency described above, a daytime maximum concentration of OH of $4 \times 10^6$ $cm^{-3}$ would increase the HONO limit of detection by approximately 20% (10 min average).

L283: Typical accuracies for the determination of OH concentration in a calibration source like used in this work are within the range of 10 to 20% and would significantly contribute to the overall uncertainty of the HONO calibration. Please clarify.

*The estimated uncertainty of 35% for the HONO measurement includes the uncertainty associated with the OH measurement (18%). This has been clarified on page 9 of the revised manuscript.*

The overall calibration uncertainty is estimated to be 35% ($1\sigma$), including the uncertainty associated with the OH calibration (18%, $1\sigma$), and depends on the precision of the photofragmentation efficiency measurement.

---

## Author Comment (AC2)

We would like to thank the reviewers for their efforts in reviewing this manuscript, and we feel that the manuscript is much stronger with their suggested changes. Below are detailed responses to their comments, which are highlighted in italics.

Reviewer #2

The authors report the development, characterisation and initial results from a novel instrument designed to detect HONO and OH in the atmosphere. The instrument is based on the fluorescence assay by gas expansion (FAGE) technique used to detect OH radicals in the atmosphere by laser induced fluorescence, which has been developed to provide measurements of HONO by measuring the OH fragment produced following the laser photolysis of HONO.

The manuscript is well presented and provides a detailed description of the instrument, its calibration, and initial results obtained in both outdoor and indoor field measurements. The description of the development and calibration of the instrument are within the scope of the journal, with the field measurements providing an indication of the capabilities of the instrument and as such are relevant to the publication.

There are, however, a number of areas in which the manuscript could be improved prior to publication which are listed below:

Line 26: 'hydroperoxy' is generally preferred over 'hydroperoxyl'.

*This has been corrected.*

Line 37: Although likely clear to most readers, the terms in equation 2 ought to be defined for clarity.

*We have defined the terms in this equation as suggested.*

Line 40: It may also be worth commenting that the wavelengths at which HONO photolyses to produce OH compared to other OH sources contribute to its role as a significant early morning OH source.

*We have added a statement reflecting this on page 2 of the revised manuscript as suggested:*

Compared to other photolytic sources of OH, the longer wavelengths at which HONO photolyzes to produce OH can result in HONO photolysis dominating OH production during the morning hours in some environments.

Line 95 onwards: There have been reports of photofragmentation-LIF technique used to measure HONO in laboratory experiments (e.g. Dyson et al., 2021 doi.org/10.5194/acp-21-5755-2021) which should be included in the discussion.

*We have added the Dyson et al. (2021) reference as suggested on page 3 of the revised manuscript, as well as a comparison of the reported limit of detection to that in this paper on page 9 of the revised manuscript:*

*Page 3:*

More recently, Dyson et al. (2021) report the detection of HONO in a laboratory setting using laser-photolysis of HONO at 355 nm and subsequent detection of OH at 308 nm in a low-pressure detection cell, reporting a detection limit of 12 ppt for a 50-s average.

Page 9:

The limit of detection for HONO described above is similar to that reported by Dyson et al. (2021) in a laboratory setting using a similar instrument employing a 355 nm laser operating at 10 Hz for photofragmentation of HONO, and detection of OH at 308 nm using a dye laser operating at 5 kHz (12 ppt, 50-s average). While the details of the photofragmentation laser in this study were not provided, the lower repetition rate of the 355 nm laser likely leads to a higher pulse energy and a higher photofragmentation efficiency compared to the 10 kHz photofragmentation laser employed in this study. However, the higher pulse energy could lead to photolysis of other ambient species that could produce OH and interfere with the measurements of HONO (see below). However, these potential interferences can be minimized in a laboratory setting.

Line 102: The wavelength limit for O($^1$D) production from ozone photolysis is usually given as 340 nm. Please clarify.

*This has been corrected.*

Line 178 (and elsewhere): 'Criegee intermediates' is preferred over 'Criegee radicals'.

*This has been corrected.*

Lines 180-183: It is not entirely clear from the description whether this method is used in any of the HONO measurements reported to remove ambient OH. Please clarify.

*We have clarified that the chemical modulation technique was not used in the HONO measurements reported in this study on pages 5-6 of the revised manuscript.*

While the LP/LIF instrument can incorporate chemical modulation cycles to measure potential interferences, the technique was not used in the HONO and OH measurements reported below in order to increase the HONO measurement frequency.

Line 195: Add the physical state for HONO.

*This has been clarified.*

Line 196: What are typical concentrations generated by this method?

*We have clarified that the method can produce a wide range of mixing ratios up to 20 ppm on page 6 of the revised manuscript.*

This method requires the reaction chamber to be heated to 50°C and can produce mixing ratios of HONO over a wide range (5-20000ppb) (Febo et al., 1995), that often require large dilution flows to reach typical outdoor atmospheric concentrations (Lao et al., 2020).

Line 199: How would production of ClNO result affect the measurement of HONO? Is it a problem for this technique?

*While production of ClNO can interfere with other HONO measurement techniques, it does not interfere with HONO measurements by the LP/LIF technique. This statement as been removed to avoid confusion.*

Line 201: It's not clear what the "long warmup times" refer to, what requires warming up with the method described?

*The production of HONO using the method described in Febo et al. (1995) requires the reaction chamber to be heated to 50 °C. This has clarified on page 6 of the revised manuscript.*

This method requires the reaction chamber to be heated to 50°C and can produce mixing ratios of HONO over a wide range (5-20000ppb) (Febo et al., 1995), that often require large dilution flows to reach typical outdoor atmospheric concentrations (Lao et al., 2020).

Line 221: It might be more appropriate to refer to an effective oxygen absorption cross-section rather than state the absorption cross-section is dependent on operating conditions. The cross-section itself is a fundamental physical property, it is more the measurement of the cross-section that depends on conditions.

*We have referred to the effective oxygen absorption cross section in this paragraph as suggested.*

Line 242 onwards: Is there any other chemistry that should be considered? Model simulations are referred to in line 250 but no information is provided on the model or mechanism. Some additional details are required on this, and it may be appropriate to include Figure S4 in the main text.

*The model uses the RACM2 mechanism, constrained by the concentrations of water and oxygen. This has been clarified on page 7 of the revised manuscript. We have also expanded the discussion of the model as suggested as well as in response to a comment by Reviewer #1. We have also included an updated version of Fig. S4 in the main text as Figure 5 as suggested.*

Figure 5 illustrates model simulations of the conversion of OH and $HO_2$ into HONO using the RACM2 mechanism constrained to the concentrations of water vapor and oxygen. After production of OH and $HO_2$ in the illuminated region of the calibrator (first 10 ms), reactions with NO lead to the production of HONO after the approximate 80 ms residence time inside the calibrator. In these simulations, the photolysis of water vapor is adjusted to produce approximately 1 ppb of both OH and $HO_2$ in the calibrator, which in the absence of NO decreases after illumination due to loss from radical-radical reactions and surface loss (Fig. 5). During typical OH sensitivity calibrations, measurements of the loss of radicals in the absence of NO is measured by changing the location of the light source relative to the exit of the calibrator (Dusanter et al., 2008). These measurements indicate that 20-30% of the OH and $HO_2$ radicals produced are lost due to reaction with the calibrator surfaces as well as loss due to the OH + $HO_2$ reaction. Model simulations indicate that a first order loss rate of 2.6 s$^{-1}$ is needed to match this observed loss of OH radicals in the calibrator in the absence of NO, and this loss rate has been included in the simulations (Fig. 5). However, these simulations suggest that during photolysis efficiency calibrations, the excess of NO is sufficient to ensure that reaction with NO is the dominant radical sink accounting for greater than 95% of the total loss of OH, with less than 3% of the OH radicals lost via surface reactions and less than 2% are lost by the OH + $HO_2$ and other radical-radical reactions. Model simulations of this

chemistry also suggest that after addition of NO, the OH and HO$_2$ concentrations are negligible and the concentration of HONO is nearly equal to the total OH and HO$_2$ concentrations produced by the calibrator (Fig. 5).

Lines 258 and 271: Can the authors comment on the identities and possible concentrations of the impurities in the NO leading to the production of OH or reduction in apparent photofragmentation efficiency? Could the same species result in interferences in ambient measurements? See also comment below. Are scrubbers to remove NO$_2$ used in the calibrations described in this work

*The source of the impurity in the NO leading to the production of OH by the photofragmentation laser is not clear but could be the result of heterogeneous reactions of NO$_2$ in the calibration system leading to the production of impurity HONO. Experiments employing the use of a scrubber to remove NO$_2$ from the cylinder such as FeSO$_4$·7H$_2$O, did not appear to impact the level of OH production, suggesting that production of this impurity may occur inside the NO cylinder. Additional experiments will be needed to identify this impurity. This has been clarified on page 8 of the revised manuscript.*

The source of this impurity is not clear but could be the result of heterogeneous reactions of NO2 in the calibration system leading to the production of impurity HONO. Experiments employing the use of a scrubber to remove NO$_2$ from the cylinder such as iron(II) sulfate heptahydrate (FeSO$_4$·7H$_2$O, Fisher scientific) did not appear to impact the OH signal due to the impurity, suggesting that production of this impurity may occur inside the NO cylinder. Additional experiments will be needed to identify this impurity.

*We have also expanded the discussion of potential impurities in the added NO that could reduce the apparent photofragmentation efficiency. Model simulations suggest that a 5% NO$_2$ impurity could reduce the production of HONO by approximately 10% due to reaction of OH with NO$_2$ instead of NO. This has been clarified on page 8 of the revised manuscript and illustrated in the new Fig. 5.*

Impurities in the added NO that react quickly with OH and compete with reaction of NO, such as NO$_2$, could lead to apparent lower photofragmentation efficiencies by reducing the amount of HONO produced in the calibrator. Model simulations suggest that a 5% NO$_2$ impurity could reduce the production of HONO by approximately 10% due to reaction of OH with NO$_2$ instead of NO (Fig, 5). As a result, the NO added should be of high purity, and chemical scrubbers designed to reduce impurity NO$_2$ should be used.

Line 285: Are there any measurements to show how much (or how little) the photofragmentation efficiency varies throughout a prolonged measurement period? A table summarising detection limits, photofragmentation efficiencies, uncertainties, conditions etc. for calibration experiments and the measurements reported in sections 3.1 and 3.2 would be useful.

*During the HOMEChem campaign, an intercomparison of the LP/LIF measurements with the University of Toronto CIMS instrument were found to agree to within ±20% over the entire campaign. These results suggest that variations of the photofragmentation efficiency over the entire month-long campaign were less than the overall instrumental uncertainty reported here. This has been clarified on page 9 of the revised manuscript. We have also included examples of the intercomparison of the LP/LIF instrument with the University of Toronto CIMS instrument during the HOMEChem campaign in Figure 9 of the revised manuscript. As suggested, we have included a table summarizing the detection limits, etc. in the Supplementary Material (Table S1).*

As mentioned above, the large uncertainty associated with the photofragmentation measurements is likely due to shifts in the overlap between the two laser beams as a result of temperature fluctuations impacting the optical alignment. Although this uncertainty is currently large, measurements of HONO were in good agreement with an acetate CIMS instrument during the recent HOMEChem (House Observations of Microbial and Environmental Chemistry) indoor measurement campaign (Wang et al., 2020a). Overall, the measurements of HONO by the LP/LIF instrument agreed with the CIMS measurements to within ±20%, on average, suggesting that variations of the photofragmentation efficiency over the entire month-long campaign were less than the overall instrumental uncertainty reported here. An example of the measurements during the intercomparison is illustrated below, and a detailed analysis of the intercomparison, including an analysis of the spatial distribution of indoor HONO emissions, will be presented in a future publication.

Line 306: What is the estimate based on?

*The mixing ratio of NO for the measurements at the forest site was based on previous measurements of the NO$_2$/NO ratio at this site. This has been clarified on page 10 of the revised manuscript.*

At this site, average mixing ratios of NO2 varied from less than 500 ppt at night up to approximately 1 ppb during morning rush hour while mixing ratios of NO were below the detection limit of the instrument but estimated to be less than 300 ppt based on previous measurements of the NO$_2$/NO ratio at this site (Lew et al., 2020).

Line 380 onwards: The manuscript would benefit from a more detailed description of potential interferences. Previous work by Liao et al. is referenced, but a more complete discussion of this work and its relevance to the current work is required. It would help to show some model calculations of species that may photolyse at 355 nm and lead to OH signals to demonstrate the impact of potential interferences. HOCl is mentioned as a potential problem during the indoor measurements, it would help if the authors could present some model calculations to indicate whether this is likely only a problem for indoor measurements, or whether ambient concentrations of HOCl could be problematic in outdoor field measurements. Similarly, are there any potential issues relating to HOBr or HOI in marine environments? Are there are other species which might significantly impact the measurement through production of OH or a reduction in the effective photofragmentation efficiency? Does absorption of the 355 nm light by other ambient species, such as formaldehyde, cause any potential interference by reducing the effective photofragmentation efficiency of HONO?

*We have expanded the discussion of interferences as suggested, including an expanded discussion of the previous work by Liao et al. (2006a). We have also included an example of an interference during HOMEChem bleach mopping events that is likely due to elevated concentrations of HOCl produced by the event (Fig. 9). We have also included a discussion of the potential interference from ambient concentrations of HOBr and HOI as suggested. The discussion also includes an illustration of the agreement of the HONO measurements by the LP/LIF instrument with that from the University of Toronto CIMS instrument during a HOMEChem cooking episode, which likely elevated indoor concentrations of formaldehyde. The agreement between the two instruments during this event suggests that photolysis of elevated indoor concentrations of formaldehyde does not significantly interfere with the LP/LIF measurements. In addition, the agreement suggests that absorption of the 355 nm light by formaldehyde from the photofragmentation laser does not significantly impact the photofragmentation efficiency. However, additional measurements together with quantification of*

*formaldehyde concentrations will be needed to confirm these results. This has been clarified on pages 12-13 of the revised manuscript.*

[revised manuscript text omitted]

Line 412 onwards: The conclusions section reads more as a section on future work. It would help to include a summary of the operating conditions and measurement capabilities.

*We have revised the conclusion section and included a summary of the operating conditions and measurement capabilities on page 13 of the revised manuscript as suggested.*

The LP/LIF instrument described here demonstrates a sensitivity and limit of detection for HONO that is sufficient for ambient measurements of HONO in both indoor and outdoor environments. Incorporating two separate lasers and employing excitation and detection of the OH fragment at 308 nm in addition to ambient sampling at low pressure minimizes interferences from laser generated OH that may have impacted previous LP/LIF measurements of HONO at atmospheric pressure (Liao et al., 2006a). The LP/LIF instrument has a $1\sigma$ detection limit for HONO of approximately 9 ppt for a 10-min integration time using 1.5W of radiation at 355 nm and a repetition rate of 10 kHz for photofragmentation of HONO, and 1-3 mW at 308 nm and a repetition rate of 10 kHz for detection of the OH fragment. The instrument is calibrated by determining the photofragmentation efficiency of HONO and sensitivity to detection of the OH fragment through the titration of a known concentration of OH from the photo-dissociation of water vapor with nitric oxide to produce a known concentration of HONO. Measurement of the concentration of the OH radical fragment relative to the concentration of HONO provides a measurement of the photofragmentation efficiency. The overall calibration uncertainty is estimated to be 35% ($1\sigma$), including the uncertainty associated with the OH calibration (18%, $1\sigma$), and depends on the precision of the photofragmentation efficiency measurement. The current limit of detection of HONO can be improved by increasing the photofragmentation efficiency with a more powerful laser system, increasing the OH detection efficiency by increasing the laser power at 308 nm, and through improvements to fluorescence detection efficiency and the overlap of the photofragmentation and excitation lasers.

Figure 3: The image quality may need to be improved prior to final publication.

*We have improved the image quality for publication as suggested.*

[Figure]

Figure 3: Timing schematic depicting one photofragmentation/excitation/detection cycle.

Figure 4: Can the authors clarify what is meant by 'S1,3', is this the average of S1 and S3?

*S1,3 is the average background signal of S1 and S3 ($S_{bkg}$). This has been corrected in the figure and clarified in the caption and in the text on page 5 of the revised manuscript.*

The net HONO signal is obtained from the difference between the signals from cycles 2 and 4 ($Net_{HONO} = S_2 - S_4$), while the net ambient OH signal ($Net_{OH}$) is obtained from the difference between the signals from cycles 4 and the background signal ($S_{bkg}$) which is the average of cycles 3 and 1 ($Net_{OH} = S_4 - S_{bkg}$).

[Figure]

Figure 4: Sample measurement cycle from the LP/LIF instrument during measurement of OH and HONO during laboratory calibrations. The raw signal has not been normalized for the power of the excitation laser or the total radical concentration produced by the calibrator. $S_{bkg}$ is the average of the offline signals $S_1$ and $S_3$

Figure 5: Can the symbols be matched to those given in Figure 4?

*We have clarified the symbols in both figures as they are different. Figure 4 represents a typical measurement cycle to measure the signal due to HONO, the signal due to OH, and the background signal. Figure 5 (now Figure 6) represents a photofragmentation calibration experiment, where the signals represent the measured total OH signal produced by the calibrator before NO is added ($S_{initial\ OH}$), the remaining OH signal when NO is added ($S_{remaining\ OH}$), the signal due to OH produced by the photofragmentation of HONO ($S_{HONO+remaining\ OH}$), and the signal to impurities ($S_{impurity}$). Since the symbols represent different measurements, we have revised the symbols in the figures, captions, and the text on page 8 of the revised manuscript so that they are different.*

Figure 6 illustrates a typical measurement of the photofragmentation efficiency. The original signal from the initial amount of OH produced in the calibrator in the absence of added NO is shown in panel (a) ($S_{initial\ OH}$), and the remaining OH concentration after NO is added to the calibrator is shown in panel (b) with the 355-nm photofragmentation laser blocked from entering the detection cell ($S_{remaining\ OH}$). This remaining OH signal is likely due to reactant segregation in the turbulent flow of the calibrator preventing all of the OH from reacting with the added NO. When the 355-nm photofragmentation laser is turned on, the increase in the signal relative to the remaining OH reflects the additional OH produced in the detection cell from HONO photolysis ($S_{HONO+remaining\ OH}$) (Fig. 6c).

[Figure]

Figure 6: Example of photofragmentation efficiency measurements from an OH + NO → HONO calibration. (a) The signal observed from total amount of OH produced in the calibrator ($S_{total\ OH}$). (b) Signal observed from remaining OH after the addition of NO converts the majority of radicals to HONO ($S_{remaining\ OH}$). (c) Sum of signal from remaining OH and signal from OH produced in the detection cell after HONO photolysis ($S_{HONO+remaining\ OH}$). (d) Signal from the 355-nm photolysis of impurities in the NO cylinder observed when the radical source in the calibrator is turned off ($S_{impurity}$).

Figure S1: Please clarify whether the laser used to photolyse HONO was a Nd:YAG (355 nm third harmonic) or Nd:YLF (351 nm third harmonic).

*We have clarified that the laser used is a Nd:YAG with 355 nm third harmonic.*

Figure S2: A schematic diagram may be more helpful than the figure provided.

*We have replaced Figure S2 with a schematic diagram as suggested.*

[Figure]

Figure S2: Schematic diagram of the water vapor photolysis calibration source.

---

## Author Response (AR2)

Dear Lisa,

Thank you for considering our manuscript. As suggested, we have shortened sections 3.1 and 3.2 by removing several paragraphs and sentences that compared our measured mixing ratios of HONO to other measurements in similar environments. We have left the discussion of the measured and modeled HONO/OH ratio, as we feel that it demonstrates an advantage of the LP/LIF instrument over other techniques.

We hope that with these changes that the manuscript will be suitable for publication in AMT.

Best regards,

Phil

The following has been removed from Section 3.1:

[revised manuscript text omitted]